# Systematic functional analysis of *Leishmania* protein kinases identifies regulators of differentiation or survival

N. Baker[1,2,6], C. M. C. Catta-Preta [1,2,6], R. Neish [1,2], J. Sadlova [3], B. Powell [4], E. V. C. Alves-Ferreira[1,2], V. Geoghegan[1,2], J. B. T. Carnielli [1,2], K. Newling [2], C. Hughes[1,2], B. Vojtkova[3], J. Anand[1,2], A. Mihut[2], P. B. Walrad [1,2], L. G. Wilson [1,5], J. W. Pitchford[2,4], P. Volf [3] & J. C. Mottram [1,2✉]

Differentiation between distinct stages is fundamental for the life cycle of intracellular protozoan parasites and for transmission between hosts, requiring stringent spatial and temporal regulation. Here, we apply kinome-wide gene deletion and gene tagging in *Leishmania mexicana* promastigotes to define protein kinases with life cycle transition roles. Whilst 162 are dispensable, 44 protein kinase genes are refractory to deletion in promastigotes and are likely core genes required for parasite replication. Phenotyping of pooled gene deletion mutants using bar-seq and projection pursuit clustering reveal functional phenotypic groups of protein kinases involved in differentiation from metacyclic promastigote to amastigote, growth and survival in macrophages and mice, colonisation of the sand fly and motility. This unbiased interrogation of protein kinase function in *Leishmania* allows targeted investigation of organelle-associated signalling pathways required for successful intracellular parasitism.

---

[1] York Biomedical Research Institute, University of York, York, UK. [2] Department of Biology, University of York, York, UK. [3] Department of Parasitology, Faculty of Science, Charles University, Prague, Czech Republic. [4] Department of Mathematics, University of York, York, UK. [5] Department of Physics, University of York, York, UK. [6] These authors contributed equally: N. Baker, C.M.C. Catta-Preta. ✉email: jeremy.mottram@york.ac.uk

The life cycles of many eukaryotic microbial pathogens are intricate due to the requirement to transition between insect and mammalian hosts and cell type differentiation is central to their ability to adapt to different environments[1]. Some parasitic protozoa undergo cell cycle arrest in response to parasite-derived autocrine signals in their host. These arrested cells can undergo differentiation in response to environmental cues to produce a distinct cell type that can proliferate. Little is known about the molecular mechanisms that underpin these events, although recent advances have been made in the malaria parasite *Plasmodium*[2,3] and the African trypanosome *Trypanosoma brucei* implicating protein kinase signalling pathways[4,5]. As models to investigate life cycle progression *Plasmodium* is relevant for its mosquito transmission and intra-erythrocytic replication, whilst *Trypanosoma brucei* is pertinent for its transmission via the tsetse fly and extracellular replication. *Leishmania*, a kinetoplastid parasitic protozoa that is the causative agent of leishmaniasis, also has a complex life cycle with a requirement to replicate and differentiate in both mammalian and sand fly (Diptera: Psychodidae) hosts. The parasite is transmitted during the vector blood meal on an infected vertebrate with the ingestion of infected cells or amastigote forms that have reached the extracellular milieu in the mammalian host. In the vector midgut, the protozoa are enclosed in the peritrophic matrix, in contact with the blood to be digested, and differentiate into procyclic promastigotes. After blood digestion, the parasites attach to midgut epithelium, replicate, differentiate and migrate to reach the anterior part of the thoracic midgut and colonise the stomodeal valve. This localisation of parasites is a prerequisite to successful transmission[6]. The infective stage is the metacyclic promastigote, adapted to migration, transmission and invasion[7,8]. After entering the mammalian host *Leishmania* metacyclic promastigotes are taken up by phagocytes, where they survive and proliferate in the acidic environment of the parasitophorous vacuole. In vitro systems are available for most life cycle stages of *Leishmania*, although these models do not completely reflect the natural environment and may not activate some of the intrinsic pathways required for normal differentiation.

Phosphorylation-mediated signal transduction in eukaryotes is a key regulator of protein function and likely has a pivotal role in *Leishmania* differentiation considering the changes reported for protein phosphorylation in different life cycle stages[9]. *Leishmania* has 195 eukaryotic protein kinases (ePKs) divided into six groups, classified according to catalytic domain conservation (CMGC, AGC, CAMK, STE, NEK, CK1), and an additional group here classified as 'Others', which contains diverse important eukaryotic cell signal transduction functions such as Aurora, Polo and AMP kinases. Atypical protein kinases (aPKs) lacking a conserved ePK structure but otherwise catalytically active are present in *Leishmania* that harbours, for example, 11 phosphatidylinositol 3′ kinase-related kinases (PIKK) resembling lipid kinases[10]. The kinome of *Leishmania* is greatly reduced in comparison to the human equivalent, lacking tyrosine kinases (TK) and tyrosine kinase-like (TKL) groups, similar to *T. brucei* and *Plasmodium*[11,12]. In contrast, CMGC, STE and NEK groups are significantly expanded in *Leishmania* relative to humans.

A number of studies have explored the role of individual *Leishmania* protein kinases in promastigote survival, metacyclic differentiation and amastigote replication, as well as their role in infectivity. Less than 10% of the kinome, mostly from the CMGC group, have been investigated using genetic and chemical approaches[13–28]. There are 17 mitogen-activated protein kinases (MAPK) in *Leishmania* and they were the focus of most of those studies, which highlighted the roles of MPK9 and MPK3 on flagellum maintenance, the latter being phosphorylated by MKK1[14,15,21]. Another example is MPK2, previously

demonstrated to modulate aquaglyceroporin 1 (AQP1) and amino acid transporter (APP3), proteins involved in drug resistance and the arginine depletion response, respectively, and therefore intra-macrophage amastigote survival[18,19]. Amastigote growth suppression and persistence were observed in TOR3 and MAPK4 null mutants[22,29], MPK10 has been implicated in amastigote differentiation, and CRK1 and CRK3 in cell cycle control and amastigote survival[13,24,30]. Only DYRK1 was shown to be essential for differentiation to metacyclic promastigotes[28], and there is a gap of knowledge on the role of protein kinases in differentiation from promastigotes to amastigotes and survival in the sand fly.

Here, we systematically tag protein kinases with mNeonGreen fluorescent protein for localisation studies, and create individual null mutants for *Leishmania mexicana* ePKs and PIKK using CRISPR–Cas9. We perform pooled infections and assess the relative proportion of barcodes representing 159 *Leishmania* protein kinase mutants to establish their importance in survival, differentiation and ultimately infection success in vitro and in vivo in the invertebrate and vertebrate hosts.

## Results

### Gene deletion mutants identify non-essential protein kinases.
*Leishmania mexicana* eukaryotic protein kinases (ePK) and PIKK family atypical protein kinases (aPKs) gene sequences were retrieved from TriTrypDB after searching for orthologues of *L. major* and *L. infantum* genes that had been identified previously[10] and by searching for protein kinase Pfam domains. We investigated a total of 204 *L. mexicana* protein kinases, of which 193 were ePKs, distributed into 8 groups, and 11 of which were aPK PIKKs (Fig. 1a and Supplementary Table 1). We also investigated the regulatory subunits of AMPK (AMPK β and AMPKγ). The *Leishmania* kinome encodes for orthologs and paralogs across a selection of trypanosomatids. 174 protein kinases have orthologues in trypanosomes and *Leishmania*, whilst 17 protein kinases are only found in the *Leishmaniinae* (termed *Leishmaniinae* unique kinases (LUKs)). Only one protein kinase, LmxM.03.0780 was found in both *Leishmania* and *Endotrypanum*, but not *Leptomonas* or *Crithidia* (Fig. 1b and Supplementary Data 1).

The requirement for these protein kinases in promastigote survival was assessed by the attempted generation of gene deletion lines using CRISPR–Cas9. Briefly, two repair templates containing drug resistance markers (*BSD* and *PUR*) were amplified using long primers with 30 bp homology arms containing a unique 12 bp barcode and two guides[31]. The repair templates and guides were transfected into *L. mexicana* procyclic promastigotes expressing Cas9 and T7 RNA polymerase (Fig. 2a). PCR amplification was used to determine the presence of the coding sequence (CDS) and the integration of the drug resistance cassette (Fig. 2b and Supplementary Data 2). PCR confirmation of an exemplar gene deletion mutant, *Δ25.0853*, showed a PCR product for the integration of the drug resistance cassette in the specific locus, and an absence of the CDS (Fig 2b-i). In contrast, when a gene deletion mutant could not be generated, for example with *PKAC1*, either no populations were recovered after transfection, or cell lines had both a CDS and an integration PCR product (designated *PKAc1::PKAC1*, Fig. 2b-ii). This may be due to the gene or chromosome duplication, which is a common mechanism by which *Leishmania* retains an essential gene following attempts at generating gene deletion mutants[32,33]. Whole-genome sequencing of 17 mutant cell lines (15 gene deletion mutants and 2 that retained an additional CDS) in comparison with Cas9T7 confirmed the loss of reads over the CDS of interest and presence of resistance markers in this location (Fig. 2c). Of note, the copy number of some other

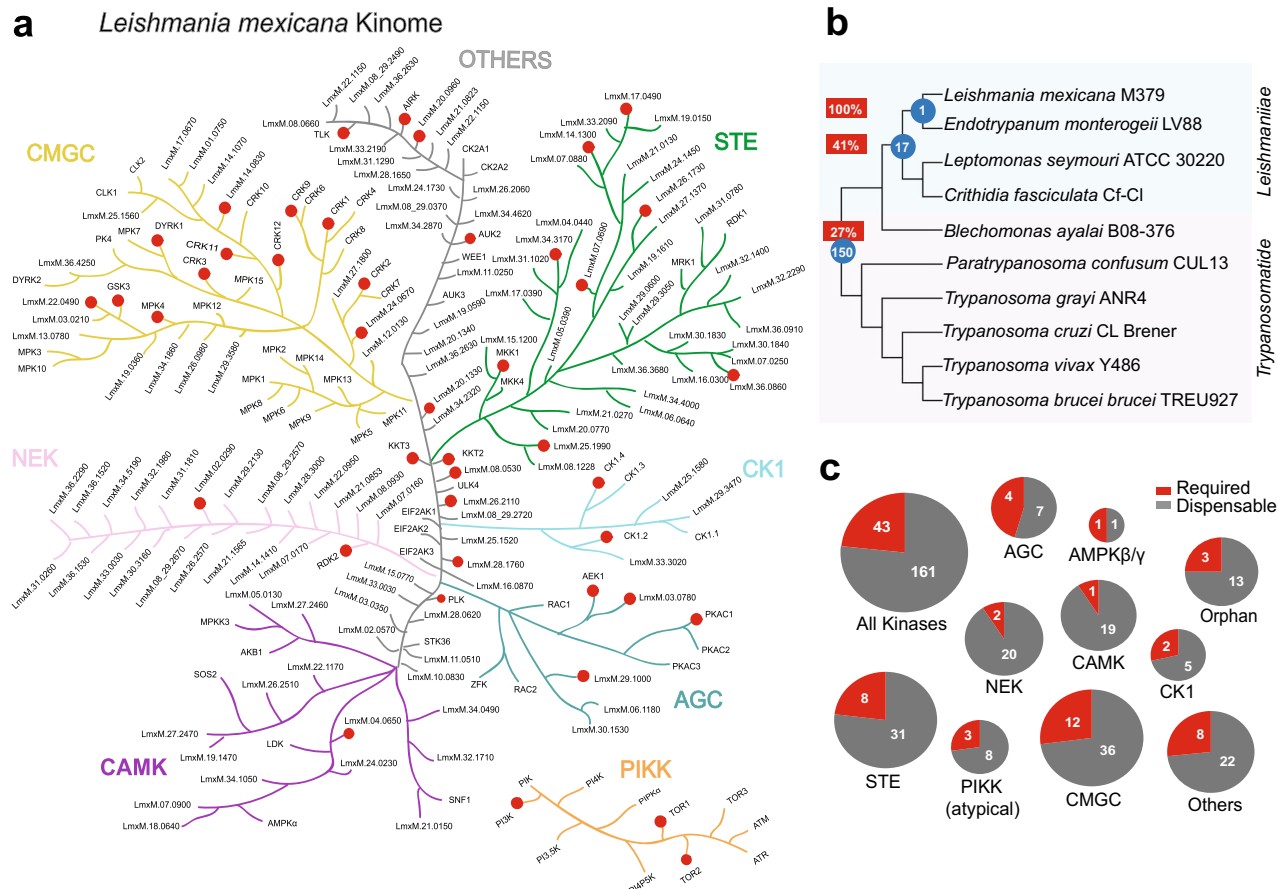

**Fig. 1 _L. mexicana_ kinome and conservation in other pathogenic trypanosomatids. a** Representation of eukaryotic (ePK) and phosphatidylinositol 3' kinase-related kinases (PIKK) protein kinase families, separated by colours defined by primary protein sequence conservation. Image for illustrative purpose only with no phylogenetic significance. Red dots represent protein kinases for which gene deletion mutants could not be generated and are therefore potentially essential genes in procyclic promastigotes. **b** Phylogenetic tree of trypanosomatids with the number of orthologous protein kinases unique to each node indicated by blue circles and the percentage of potentially essential orthologous protein kinases unique to each node indicated within a red rectangle. **c** Pie-charts show dispensable (non-essential) and required (potentially essential) protein kinases in procyclic promastigotes, separated into families.

protein kinase genes changed in the gene deletion mutants. For example, Δ06.1220 has duplicated its genome and in Δ20.1340 the copy number of _LmxM.29.0600_ and _LmxM.07.0900_ has increased; gene copy number changes could influence the expression of one protein kinase as a way to compensate for the loss of another. If a gene deletion mutant was unable to be identified after three transfection attempts, it was classed as 'required' with a 1-star quality classification[34]. Forty-three protein kinases and AMPKγ fell into this category (Fig. 1a, c).

Gene deletion mutants were successfully produced for 161 protein kinases, equating to 79% of the kinome, and the AMPK β regulatory subunit. Only 2/22 (9%) NEKs were found to be refractory to gene deletion compared to 12/48 (25%) CMCGs and 4/11 AGCs (36%) (Fig. 1c). Gene deletion mutants were obtained for all 11 pseudokinases that lack all three of the key conserved residues thought to be important for catalytic activity (Supplementary Fig. 1)[10], demonstrating no essential requirement in promastigotes. A further 40 protein kinases were found to lack either one or two of these residues and gene deletion mutants could be generated for 33/40 (82.5%). As previous work showed that it is possible to mutate the lysine residue and retain some ATP binding activity[35], we have classified these protein kinases as putatively active.

The essentiality of protein kinases was considered in relation to orthology across 10 trypanosomatid species. Of the 174 protein kinases found in all 10 of these species, 21% were found to be required in promastigotes. In contrast, 41% of the protein kinases found only in the _Leishmaniinae_ group (LUKs) were required (Fig. 1b). This suggests that _Leishmania_ promastigotes require these LUKs specifically for their life cycle adaptations. Of note, two of the eleven non-essential pseudokinases are also LUKs.

Our gene deletion attempts indicate that 43 of the protein kinases are essential in promastigotes (Table 1). For additional validation, we generated gene deletions for both endogenous alleles of _LmxM.04.0650_ whilst providing an extrachromosomal copy of the gene. Δ04.0650 [6xmyc-04.0650] (Fig. 2d) retained the episome for at least seven generations without drug pressure. In some instances, genes were refractory to the anticipated genetic manipulation and in the case of _CLK1_ (_LmxM.09.0400_) and _CLK2_ (_LmxM.09.0410_) this was found to be due to incorrect assembly of the reference genome. A recent nanopore genome allowed the design of new primers and the successful generation of Δclk1 and Δclk2 gene deletion mutants, suggesting that disruption to one of the two _CLK_ genes can be compensated for by the other (Supplementary Fig. 2).

**Localisation of protein kinases**. We generated 199 N- or C-terminal mNeonGreen tagged protein kinases for localisation studies in procyclic promastigotes (Fig. 3a). We used the atlas of _Leishmania_ cellular landmarks[36] to localise proteins to the

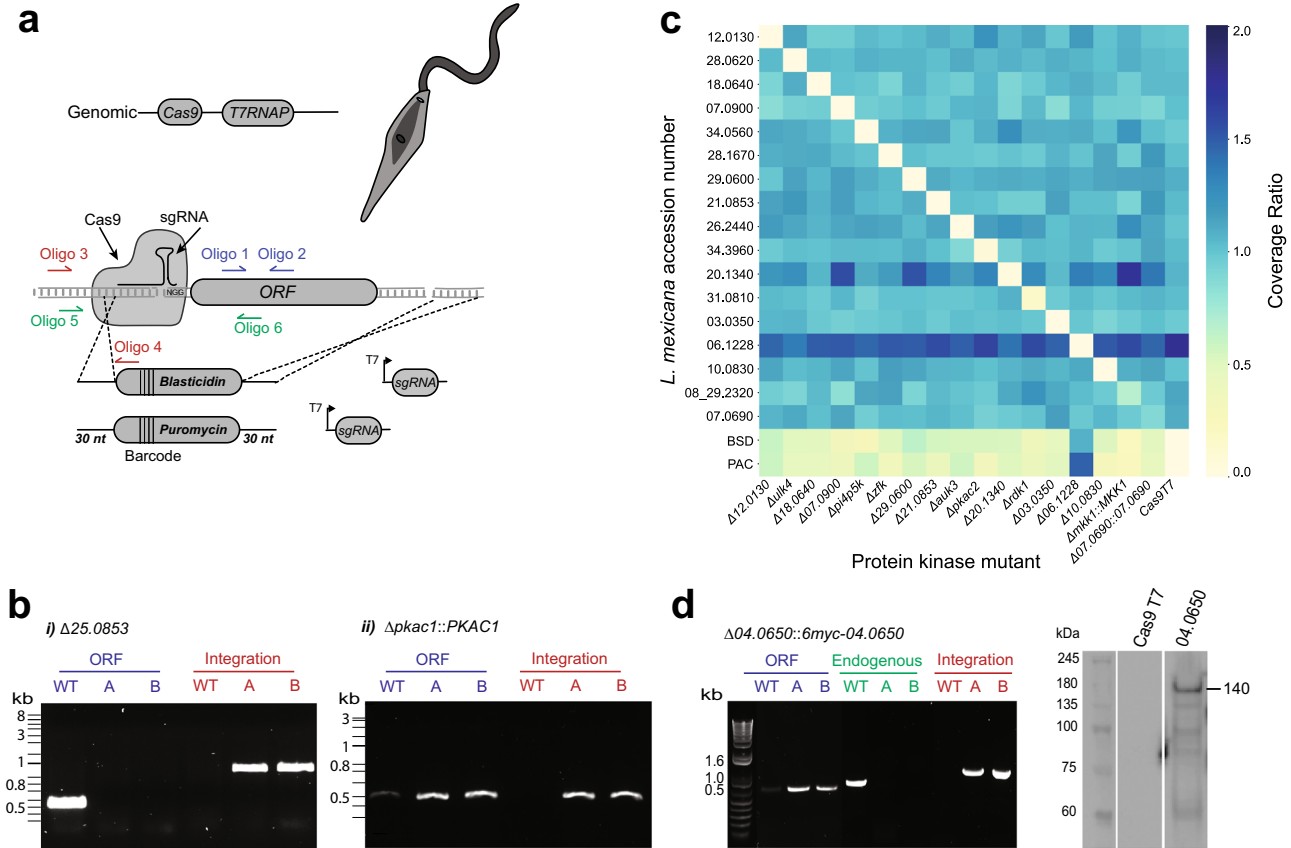

**Fig. 2 Generation of the *L. mexicana* kinome gene deletion library. a** Schematic representation of gene editing by CRISPR–Cas9 in the *L. mexicana* progenitor cell line showing the integration of repair cassettes containing 30nt homology sites, a unique barcode and antibiotic-resistance genes for puromycin (*PAC*) and blasticidin (*BSD*). Diagnostic PCRs for the presence of the coding sequence (CDS) with Oligo 1 and Oligo 2, (depicted in blue), correct integration of the repair cassettes using Oligo 3 and Oligo 4 (depicted in red), and the gene in the endogenous locus with Oligo 5 and Oligo 6 (depicted in green). **b** Diagnostic PCR for (i) *LmxM.25.0853* and (ii) *PKAC1* (*LmxM34.4010*) using primer pairs as above in *L. mexicana* Cas9T7 (WT) and Populations A and B. **c** Heat map indicating the presence or absence of target genes after whole-genome sequencing. The ratio of gene coverage to chromosome coverage was used as a measure of gene copy number in selected mutants. A value of 1 represents two alleles of a given gene. **d** Diagnostic PCR for facilitated gene deletion mutant *LmxM.04.0650* using Oligo 1 and 2 for the CDS, Oligo 5 and 6 for the endogenous locus and Oligo 3 and 4 for correct integration. Western blot using anti-myc confirmed the episomal expression of LmxM.04.0650 at the anticipated size of 140 kDa.

cytoplasm (48.5%), basal body (11.6%), nucleus (10.1%), endo-membrane (7.1%), flagellum (6.5%), lysosome (6.5%), flagellar pocket (3%), pellicular membrane (2.5%), cytoplasmic organelles (2%) and mitochondrion (1%) (Fig. 3b, c, Table 1 and Supplementary Data 3, Supplementary Data 4 and Supplementary Table 1). The fluorescence signal for some protein kinases varied during the cell cycle, for example, PKAC1 (LmxM.34.4010) accumulates in the nucleus in early G1/S and at late G2/M in the cytoplasm and flagellum. This also holds true for MPK6 (LmxM.31.3250) which is localised within the cytoplasm and flagellum in G1 but is located on the pellicular membrane during S phase. In other cases protein kinases were found in more than one location, for example, LmxM.14.1070 was found in the basal body, flagellar pocket, lysosome and cytoplasm (Supplementary Data 3).

**Protein kinases required for survival in the mammalian host.** In order to ascertain the role of protein kinases in differentiation across various life cycle stages, mutants were pooled and subjected to *Leishmania* life cycle progression. This pool (Pool 1) contained 159 barcoded mutants that included 152 protein kinase gene deletion mutants, five protein kinase mutants in which gene deletion mutants could not be generated (required), Δ*ampk β* and one non-protein kinase control, metacaspase (Δ*mca*), which was

included to provide comparator data with a previous peptidase bar-seq screen[33,37]. We tested the ability of mutants to transition through the *Leishmania* life cycle as procyclic promastigotes (PRO), metacyclic promastigotes (META), axenic amastigotes (AXA), amastigotes in macrophages (inMAC) and amastigotes in the footpads of mice (FP), taking samples at various time points for bar-seq analysis[33] (Fig. 4a). The relative growth rate of each mutant was determined by counting barcodes represented in each time point and calculating the proportion of each mutant within the population. Log-proportions were plotted as trajectories across time points for each experiment. Outputs from experimental arms 1–3 were plotted for direct comparison against one another for six example genes that had no loss of fitness, increased relative fitness or a decreased relative fitness (Fig. 4b). The eukaryotic initiation factor kinase mutant, Δ*eif2ak3* showed a strong increase in representation for axenic amastigotes, macrophage infection and infection of mice, but may not represent gain-of-fitness due to decrease in the relative abundance of mutants that have a loss of fitness. The mitogen-activated protein kinase kinase 3 (Δ*mpkk3*) and cdc2-related kinase 10 (Δ*crk10*) mutants maintained an average level of representation across all time points in all three experiments. This was in contrast to the mitogen-activated protein kinase gene deletion mutants Δ*mpk1* and Δ*mpk2*, which both demonstrated a loss of fitness in both

**Table 1 Required *L. mexicana* protein kinase genes.**

| Gene ID | Name | Group/Family | Primary localisation | Previous literature |
|---|---|---|---|---|
| LmxM.25.2340 | AEK1 | AGC | Cytoplasm | |
| LmxM.29.1000 | | AGC | Cytoplasm | |
| LmxM.34.4010 | PKAC1 | AGC/PKA | Nucleus | |
| LmxM.03.0780 | | AGC/RSK | Cytoplasm | |
| LmxM.04.0650 | | CAMK/CAMKL | Cytoplasm | |
| LmxM.27.1780 | CK1.4 | CK1/CK1 | Flagellar pocket | |
| LmxM.34.1010 | CK1.2 | CK1/CK1 | Cytoplasm | 26 |
| LmxM.24.0670 | | CMGC | Nucleus | |
| LmxM.05.0550 | CRK2 | CMGC/CDK | Cytoplasm | |
| LmxM.09.0310 | CRK12 | CMGC/CDK | Nucleus and kinetoplast | 43 |
| LmxM.21.1080 | CRK1 | CMGC/CDK | Mitochondrion | 30 |
| LmxM.27.1940 | CRK9 | CMGC/CDK | Nucleus | |
| LmxM.29.1780 | CRK11 | CMGC/CDK | Nucleus | |
| LmxM.36.0550 | CRK3 | CMGC/CDK | Cytoplasm | 24,42 |
| LmxM.14.0830 | | CMGC/DYRK | Basal body | |
| LmxM.15.0180 | DYRK1 | CMGC/DYRK | Lysosome | 28 |
| LmxM.18.0270 | GSK3 | CMGC/GSK | Pellicular membrane | 69 |
| LmxM.22.0490 | GSKA | CMGC | No signal | |
| LmxM.19.1440 | MPK4 | CMGC/MAPK | Lysosome | 14 |
| LmxM.24.2320 | MKK4 | STE | Endomembrane | |
| LmxM.25.1990 | | STE | Endomembrane | |
| LmxM.34.3170 | | STE | Cytoplasm | |
| LmxM.07.0690 | | STE | Cytoplasm | |
| LmxM.14.1300 | | STE | Flagellum | |
| LmxM.17.0490 | | STE | Flagellum | |
| LmxM.27.1370 | | STE | Endomembrane | |
| LmxM.36.0860 | MKK5 | STE | Cytoplasm | |
| LmxM.02.0290 | | Other/NEK | Cytoplasm | |
| LmxM.30.2960 | RDK2 | Other/NEK | Cytoplasm | |
| LmxM.26.2110 | | Orphan | Lysosome | |
| LmxM.34.4050 | KKT3 | Orphan | Nucleus | |
| LmxM.36.5350 | KKT2 | Orphan | Nucleus | |
| LmxM.20.1330 | | Other/unique | Endomembrane | |
| LmxM.28.0520 | AUK1/AIRK | Other/AUR | Nucleus | 70 |
| LmxM.08_29.1330 | AUK2 | Other/AUR | Cytoplasm | |
| LmxM.17.0790 | PLK | Other/PLK | Basal body | |
| LmxM.30.2860 | TLK | Other/TLK | Nucleus | |
| LmxM.28.1760 | | Other/VPS15 | Not attempted | |
| LmxM.08.0530 | | Other/unique | Endomembrane | |
| LmxM.20.0960 | | Other/unique | Endomembrane | |
| LmxM.24.2010 | PI3K | Atypical | Cytoplasm | |
| LmxM.36.6320 | TOR1 | Atypical | Cytoplasm | 22 |
| LmxM.33.4530 | TOR2 | Atypical | Cytoplasmic organelles | 22 |

Genes for which gene deletion mutants could not be generated in procyclic promastigotes and are likely to be essential. Includes previous literature about gene essentiality.

axenic and intracellular amastigote stages, a phenotype that has been reported previously[18,21,38]. Our results agree with those findings and also showed that loss of fitness is seen in the mouse footpad infection. The zinc finger kinase mutant (Δ*zfk*) also displayed a loss of fitness across the three experimental arms, but with an increased representation upon enrichment of metacyclic promastigotes.

We examined each barcode proportion profile individually (Supplementary Data 5), but also opted to use an unbiased clustering algorithm to sort the mutants into groups with similar phenotypes, taking all time points into account. We used the projection pursuit method[39] to calculate the differences between the trajectories of logged-proportions, looking at the first difference between each time point within the series, for each of the mutant strains. The projection pursuit clustering algorithm finds a 2-dimensional projection of the phenotype data in which clusters are as distinct as possible. This involves iterating between projection and clustering steps until the algorithm converges. The result is a 2-dimensional map in which whole phenotypes are represented by individual points. Clusters of points on this map were then scrutinised and checked for biological plausibility in a way that would be impractical in a higher-dimensional setting. This allowed qualitative similarities and differences between trajectories to be assessed in an unbiased way. After examining multiple solutions, we set our parameters to produce six clusters for each of the experimental arms (EA1–EA3) (Supplementary Data 5). Axenic amastigote trajectories allowed for the identification of mutants where differentiation and replication were impaired, without interference by host factors (Supplementary Data 5). Macrophage infections in vitro were used for identification of mutants for genes important during infection, either differentiation and adaptation to the acidic environment of the phagolysosome or replication inside it. The last experimental setup for in vivo infections took into account the mouse immune system, able to eliminate mutants less able to survive in this environment. The mouse footpad infection experiment, examining the first differences in the series, was therefore chosen as an example of the cluster analyses (Fig. 4c). Cluster 1 (green)

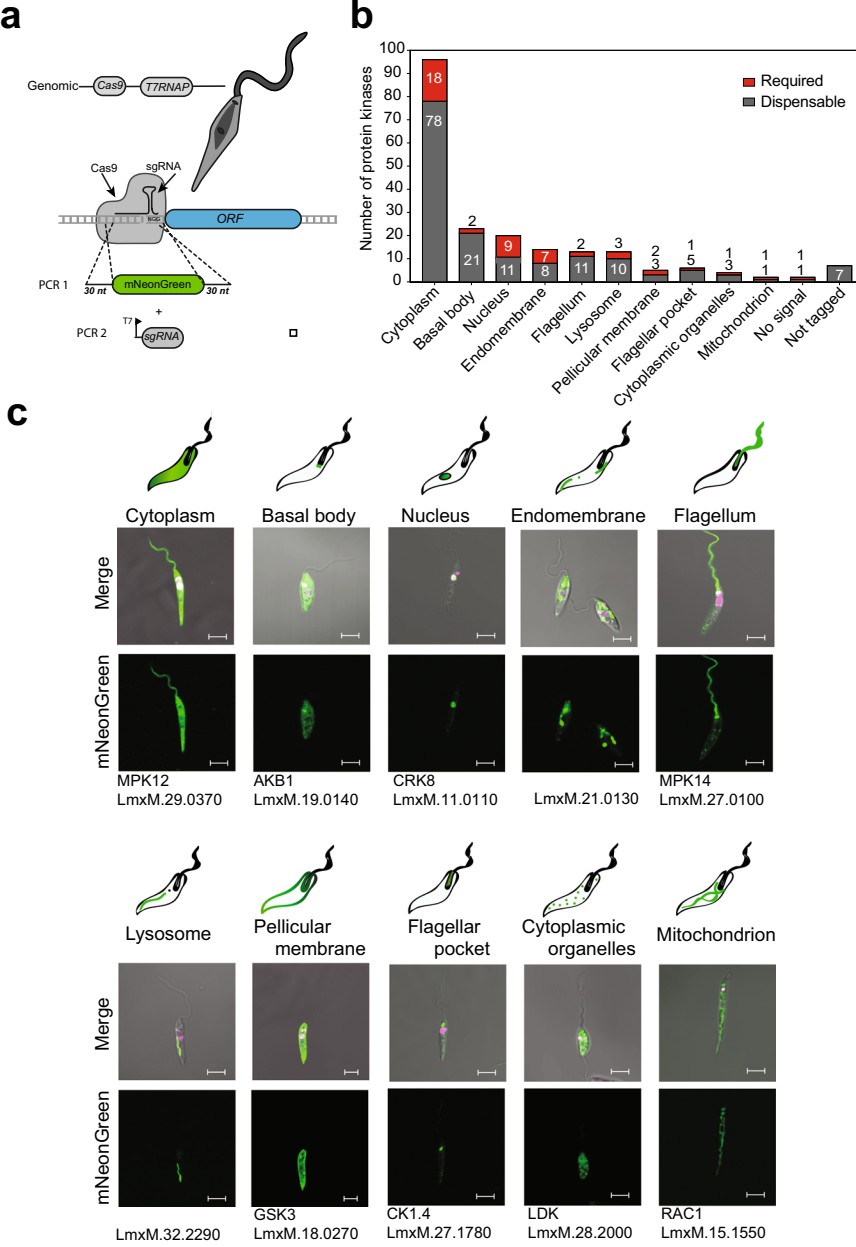

**Fig. 3 Endogenous tagging of *L. mexicana* protein kinases. a** Schematic representation of a process for N-terminal tagging. Transfection of the sgRNA (PCR1) directs the double-stranded break immediately upstream of the target CDS and the repair template (PCR2) contains 30 nt at 5′ and 3′ of the fragment for homologous recombination on the target locus to insert the fluorescent tag and an antibiotic-resistance gene. **b** Bar graph representing the localisation of dispensable and required protein kinases for each organelle. **c** Pictorial illustration of the 10 localisation groups. An exemplar live-cell image of protein kinases tagged with mNeonGreen for each localisation. Procyclic promastigotes were incubated with Hoechst 33342 for DNA labelling (pink). Scale bar: 5 μm.

contains 12 trajectories for mutants that exhibited an increased representation during the mouse infection time points compared to others, indicating a relative gain of fitness in the population. Cluster 2 (red) contains trajectories with an average representation throughout promastigote stages and small increase between the metacyclic enrichment and three-week footpad infection stage. Cluster 3 (sky blue) contains the largest group of protein kinase mutants with fairly flat trajectories, decreasing slightly on average from the point of metacyclic enrichment. Cluster 4 (purple) contains protein kinase mutants that on average exhibit fairly flat trajectories throughout the promastigote stages, demonstrating average growth, with a decrease in fitness upon footpad infection. Cluster 5 (blue) contains mutants that show a

significant increase in representation during metacyclic enrichment followed by a sharp decrease during footpad infection time points. Cluster 6 (grey) contains protein kinase mutants with trajectories which show a general loss of representation over time with a slight recovery during metacyclic enrichment. Each of the individual kinase trajectory examples (Fig. 4b and Supplementary Data 5) are found in separate clusters and provide an example of the data for the axenic amastigotes and macrophage infections. The percentage barcode representations for each of the mutants are also shown in a heat map format for each of the clusters, depicting the trends for individual mutants. PCA analysis supports the relationship between the different clusters (Fig. 4d and Supplementary Data 5).

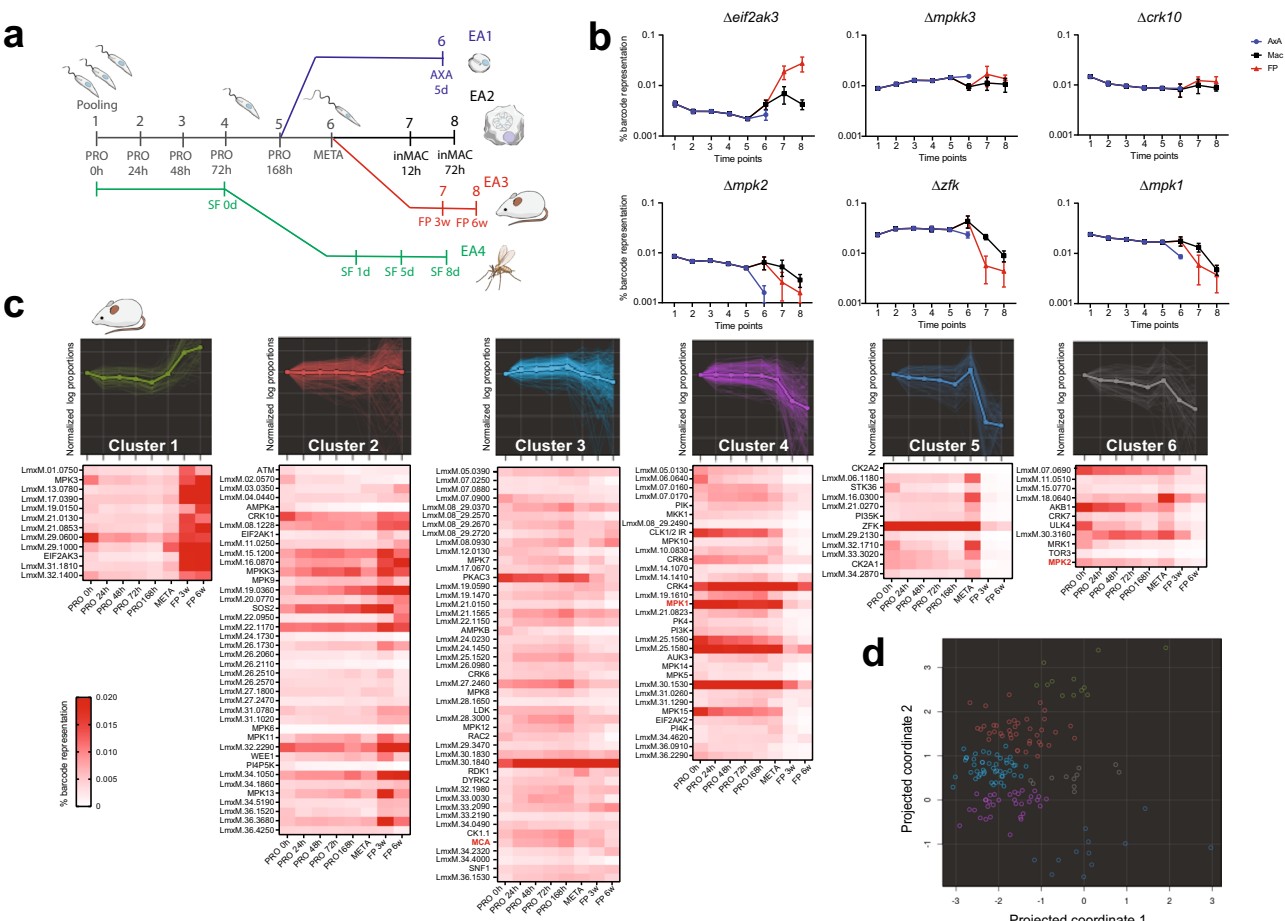

**Fig. 4 Identification of protein kinases involved in *L. mexicana* differentiation and infection. a** Schematic illustration of the bar-seq screen design with three experimental arms (EA1, blue; EA2, black; EA3, red) depicting investigation of differentiation to amastigote stages and investigation of protein kinase mutants in sand fly (SF) infection (EA4, green). A pool of gene deletion mutants was generated in procyclic promastigotes (PRO), which were grown in Graces Media at pH 5.5 for 168 h. Cells were then either diluted into amastigote media at 35 °C and grown as axenic amastigotes (AXA) for 5 d or enriched for metacyclic promastigotes. Metacyclic promastigotes (META) were used to either infect macrophages (inMAC) or inoculate mouse footpads (FP). DNA was taken at the indicated time points and the unique barcodes amplified by PCR to apply bar-seq analyses. **b** Proportion of barcodes across 8 time points, as indicated in a. The trajectories of six exemplar protein kinase mutants have been plotted for the three experimental arms. Values are mean ± S.D., n = 6 biologically independent samples. Statistical analysis was multiple *t*-test corrected for multiple comparisons with post-Holm–Sidak method. **c** Projection pursuit cluster analysis was applied to the trajectories from each experimental arm, grouped into six clusters, each consisting of trajectories with a similar trend. Only clusters resulting from the mouse footpad infection are shown. The image for each cluster shows gene trajectories overlaid with average trends in bold. Trajectories plotted using logged % barcode representation data, normalised to time 0. Heat maps below show % barcode representation data and depict the trend for each individual gene. Gene IDs in red (MCA, MPK1 and MPK2) have documented phenotypes in *Leishmania* and serve to benchmark the dataset. **d** The relationship between the clusters is shown on a two-dimensional PCA plot. Colours match the clusters in **c**.

**Protein kinases important for differentiation**. We took advantage of our three experimental arms for cross-validation of mutants showing loss of fitness in amastigote infection, differentiation and/or replication. Clusters that identified protein kinases important for the amastigote stages were compared across the three arms for a first difference output from the projection pursuit analysis. Twenty-nine protein kinase mutants were identified as important for amastigotes across at least two of the experimental arms (EA1–3) (Fig. 5a and Supplementary Data 6). These included MPK1 and MPK2 and six other protein kinase mutants that had been identified previously as being important for in vivo infection of *T. brucei*[40], as well as TOR3, previously identified as required for macrophage infections[22]. A further 15 protein kinases were identified with loss of fitness only in the mouse footpad infection (Supplementary Data 6), suggesting that these mutants can differentiate into amastigotes successfully and infect macrophages but cannot survive in the harsh in vivo environment.

We had identified previously a STE protein kinase in *T. brucei*, RDK1, as a repressor of bloodstream-form differentiation[11] and the *L. mexicana* syntenic ortholog fell into cluster 3 (Fig. 4b). We analysed the *L. mexicana* Cas9 T7 progenitor line and Δ*rdk1* for life cycle progression and found that Δ*rdk1* had the same proliferation profile as the progenitor promastigotes and axenic amastigotes in culture (Supplementary Fig. 3a, b). In contrast, Δ*rdk1* had a loss of fitness of ~40% after 72 h in the in vitro macrophage assay and a loss of fitness in mice that was complemented with the add-back, Δ*rdk1::RDK1* (Supplementary Fig. 3c, d). These analyses confirmed the bar-seq results, providing confidence and validation of the pooled library data.

**Protein kinases required for survival in the sand fly**. We next addressed the ability of gene-deficient mutants to survive in the sand fly vector *Lutzomyia longipalpis*. We put together a new pool of gene deletion mutants (Pool 2), which were separated into three sub-pools based on their relative growth rates of slow,

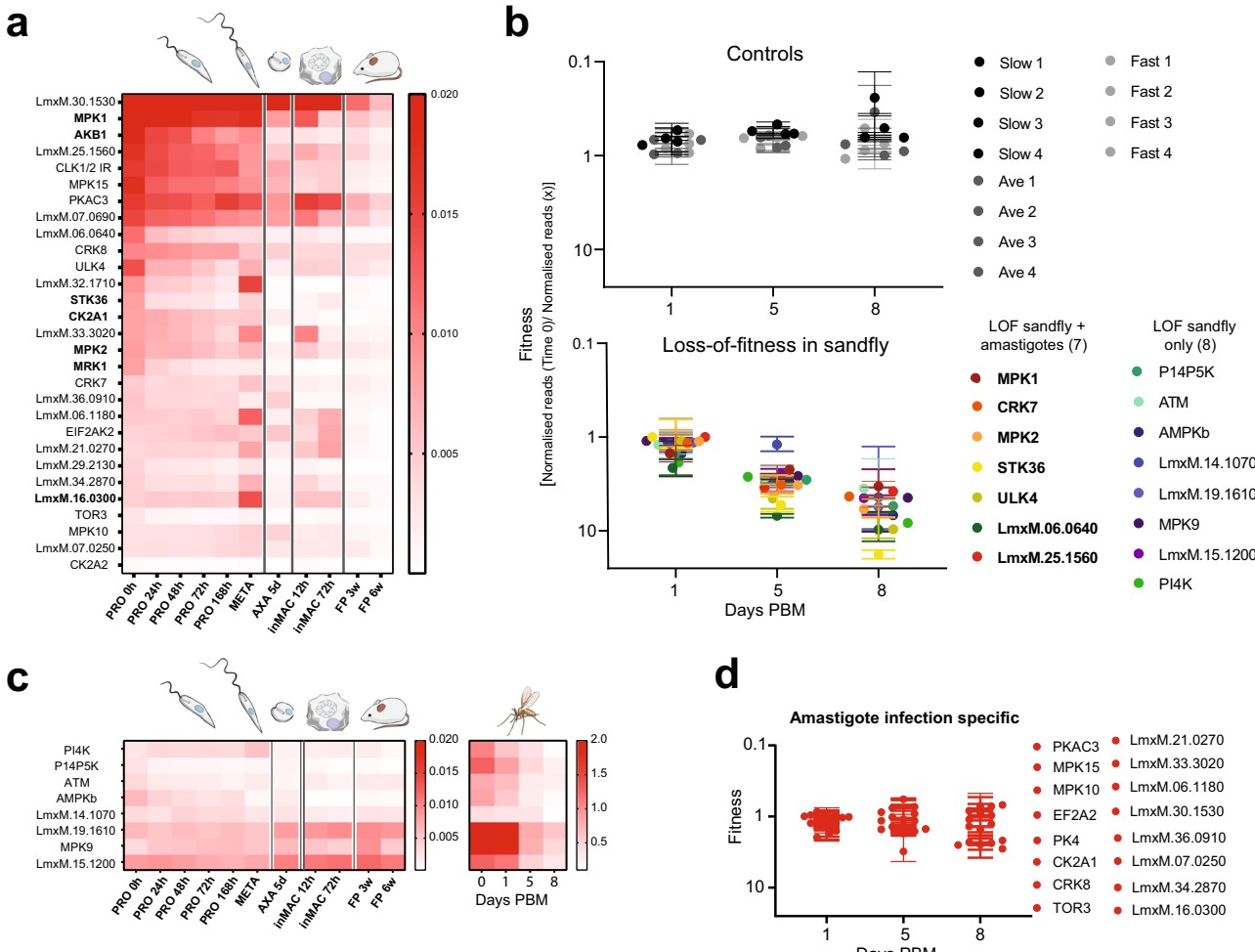

**Fig. 5 Identification of protein kinases involved in colonisation of *Lutzomyia longipalpis*. a** Heat map of 29 protein kinase mutants that cluster in groups important for the amastigote stage in two of the three experiments using 1 difference clustering. In bold are the genes that have been identified as important for mouse infection in a previous trypanosome study[40]. **b** Summary showing the protein kinases important for sand fly infection. Control plot (top) shows the relative barcode representation for four control lines with a barcode inserted in the ribosomal locus at three time points post blood meal (PBM) (days 1, 5 and 8). Protein kinases important for *L. longipalpis* infection plot (bottom) shows the 15 mutants with significant loss of representation by day 8 identified using a multiple *t*-test, two-stage linear step-up procedure of Benjamin, Krieger and Yekutieli (values are mean ± S.E., *n* = 2 biologically independent samples). 7 protein kinases identified as important for both sand fly infectivity and for survival in amastigote stages indicated in bold. **c** Heat map showing the amastigote stage data for the 8 protein kinase mutants that are important for sand fly infection only. Heat maps show the range of normalised reads. **d** Plot showing the infectivity of the protein kinase mutants important for the survival of amastigotes only, identified using multiple *t*-test corrected with post-Holm–Sidak method (values are mean ± S.E., *n* = 2 biologically independent samples).

average or fast. Each sub-pool of ~50 procyclic promastigote mutants had four additional control lines[41], where a barcode was added to the ribosomal locus (Supplementary Data 6 and Supplementary Fig. 4). These controls had been used for a previous bar-seq experiment[41] and provided a benchmark to compare sand fly infection experiments. *L. longipalpis* sand flies were allowed to feed on blood containing the three sub-pools, then bar-seq analyses were carried out on DNA samples from days 0, 1, 5 and 8 post blood meal (Supplementary Fig. 4 and Supplementary Data 6). Four mutants (*Δmpk9*, *Δ31.1020*, *Δ19.1610* and *Δ19.0360*), had a significant loss of representation after 5 days of infection, whilst fifteen protein kinase mutants had a significant loss of representation after 8 days of infection (Fig. 5b and Supplementary Data 6). The four controls maintained an average ratio throughout, as expected[41] (Fig. 5b and Supplementary Data 6).

The two in vivo screens provided a list of protein kinases important for successful infection of mice and colonisation of the

sand fly midgut. Seven protein kinases were identified as being required in both screens, indicating that they have fundamental roles and are required for differentiation and/or survival in both the insect vector and mammalian host (Fig. 5b). This list includes MPK1, MPK2, CRK7 and LmxM.06.0640, a MAP3K. Eight protein kinases, including MPK9, ATM and PI4K were identified as important for colonisation of the sand fly only, as these cluster with average trajectories in the amastigote study (Fig. 5c). Sixteen protein kinases were identified as being required for differentiation and/or survival in the mammalian host only (Fig. 5d), as these show no significant decrease in representation by day 8 post blood meal in the sand fly. These include CMGC family protein kinases MAPK10, MAPK15 and CRK8, as well as PKAC3, PK4 and TOR3.

**Protein kinases involved in flagellum function.** We hypothesised that protein kinase mutants that had defective motility as

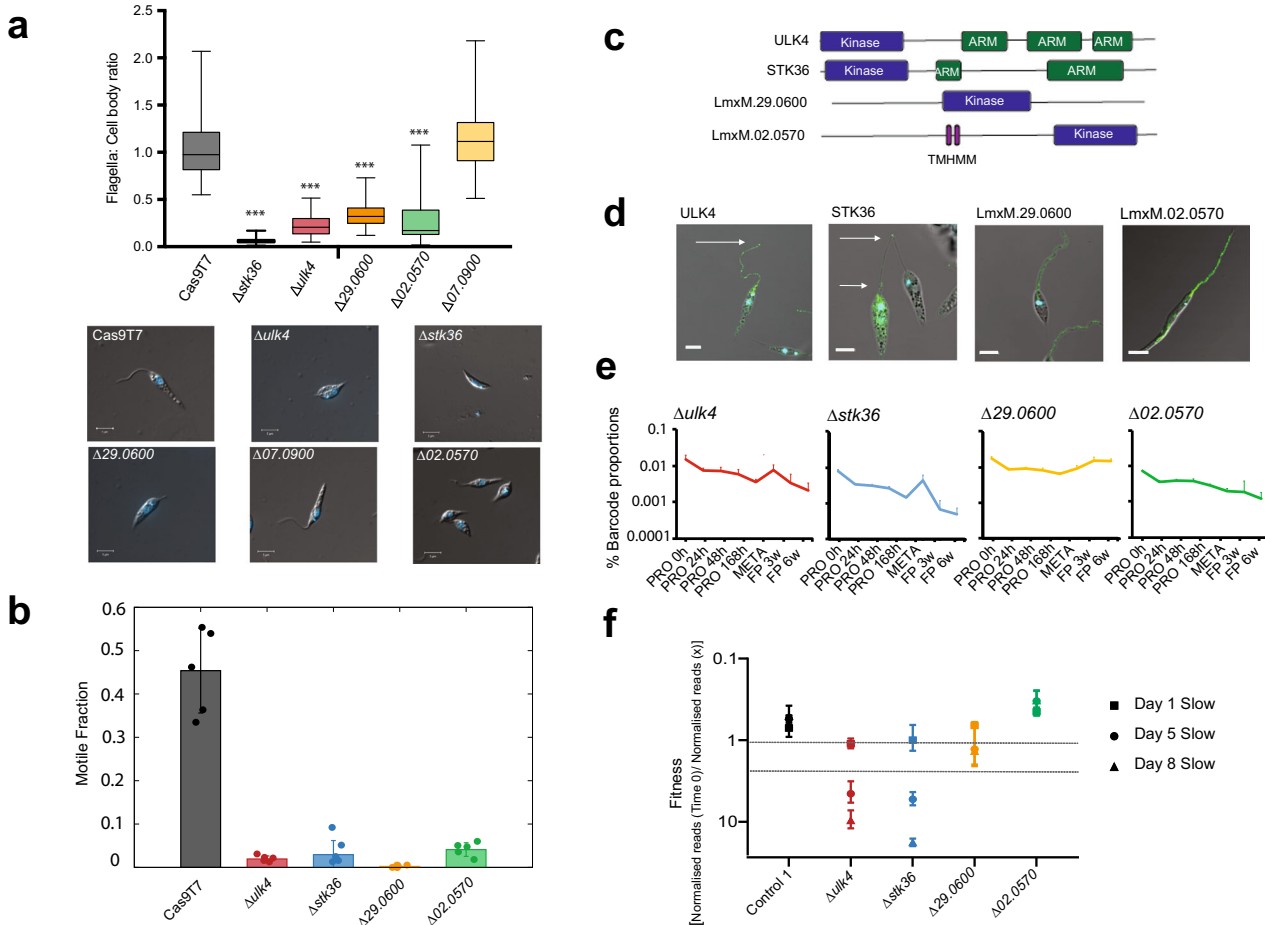

**Fig. 6 Identification of protein kinases involved in motility. a** Bar chart showing flagellar length: cell body length ratio for 5 protein kinase-deficient mutants identified with significant loss of representation at the top of the transwell, indicating a motility defect. Error bars indicate standard deviation (*n* = 50 cells for all samples except Δulk4 where there were 43 cells in the analysis). Mutants with a significant difference in flagella:cell body ratio when compared to Cas9T7 were identified using an unpaired *t*-test with post-Welch's correction (***Δulk4 *p* = 1.07e−25; Δstk36 *p* = 1.93e−27; Δ29.0600 *p* = 1.01e−22 and Δ02.0570 *p* = 1.84e−21). **b** Graph showing the fraction of the population for each promastigote protein kinase mutant progressively swimming. Five sample chambers were prepared for each cell type. The motile fraction of cells in the control (*Cas9T7*, *n* = 2532 cells) and four mutant strains: Δulk4 (*n* = 2936 cells), Δstk36 (*n* = 4896 cells), Δ29.0600 (*n* = 2623 cells) and Δ02.0570 (*n* = 3532 cells). The error bars represent standard error of the mean/95% confidence intervals and the data points denote results from individual chambers. **c** Schematic showing the protein structures of the four protein kinase mutants identified in the motility screen. **d** Localisation of the proteins tagged with mNeonGreen. **e** Bar-seq trajectories for the four protein kinase mutants for Pool 1, EA3 with loss-of-fitness analysed using multiple t-test corrected with post Holm–Sidak method (values are mean ± S.D., *n* = 6 biologically independent samples). **f** normalised bar-seq reads for these four protein kinase mutants during development in the sand fly (values are mean ± S.E., *n* = 2 biologically independent samples).

promastigotes would have a loss of fitness in the sand fly[41], but not in the mouse. To test this, we initially screened Pool 1 for mutants that had defective motility in a transwell assay. We identified 9 mutants that were significantly different between the upper and lower chambers after 4 h equilibration (Supplementary Data 6). We validated the screen with four mutants, Δulk4, Δstk36, Δ02.0570 and Δ29.0600, which had cells with significantly shortened cell body: flagellum ratio and short flagella observed by microscopy (Fig. 6a and Supplementary Fig. 5, Supplementary Data 6). These mutants also had severely impaired motility, as assessed by 2D tracking (Fig. 6b and Supplementary Movie 1). The motile cells with normal length flagella present in these populations of Δulk4 (~8%), Δstk36 (~3%) showed slow swim rates compared to the parental line (Supplementary Fig. 6). Re-expression of ULK4 in Δulk4 and STK36 in Δstk36 restored the flagellum length to that of the parental line (Supplementary Fig. 6). ULK4 and STK36 share not only similar domain architecture with an N-terminal kinase domain and a number of ARM

domains (Fig. 6c) but both are located at the flagellum and flagellar tip (Fig. 6d and Supplementary Fig. 5). Subsequently, we repeated the motility screen with Pool 2 and identified 20 protein kinases that were significantly different between the transwell chambers, 6 of which had been identified in Pool 1 (Δstk36, Δulk4, Δ29.0600, Δ36.1520, Δ02.0570 and Δ07.0900) and one of which, Δmpk3, had previously been reported to have very short flagella and reduced motility[15].

LmxM.02.0570 is one of only five *Leishmania* protein kinases with a predicted transmembrane domain and it was localised to both the lysosome and flagellum, a location found for just two other protein kinases in our dataset (LmxM.26.1730 and LmxM.36.1530). The trajectories for mouse infection for Δstk36 and Δulk4 follow a similar trend showing increased representation after enrichment of metacyclic promastigotes and decreasing representation during mouse infection and hence group with protein kinases important for amastigote differentiation and survival in clusters 5 and 6, respectively (Fig. 6e). LmxM.29.0600

and LmxM.02.0570, on the other hand, are found in clusters 1 and 2 respectively for the mouse footpad infection due to the former exhibiting a small increase in fitness and the latter a general but less significant decrease. Interestingly, STK36 and ULK4 were also identified as important protein kinases for survival in the sand fly, whereas MPK3, LmxM.29.0600 and LmxM.02.0570 were not (Fig. 6f). This suggests that these proteins are fundamental to infection for a reason independent of their short flagellum and provides evidence that not all mutants with flagella defects as promastigotes are hindered in the colonisation of the sand fly.

## Discussion

Protein kinases are fundamental to orchestrating the cellular signalling required for *Leishmania* survival throughout the life cycle. Here we have taken a holistic approach to explore the role of each protein kinase in the parasite's life cycle, first assessing whether each gene is dispensable for growth as promastigotes and then using pooled library bar-seq phenotyping to assess the ability of gene deletion mutants to progress through the life cycle. We also determined the location of each protein kinase in promastigote stage cells, allowing the generation of hypotheses for potential signalling pathways associated with specific organelles. Gene deletion mutants were successfully generated for 161 protein kinases in promastigotes, equating to 79% of the kinome, indicating a high level of functional redundancy. We were unable to generate gene deletion mutants for 43 protein kinases, which are likely to be essential for parasite proliferation. No technical difficulties could be identified with the CRISPR system, as the same guide RNAs and selection methods were used to successfully generate nNeonGreen tagged cell lines. This result is consistent with the level of essentiality for the kinome also seen using RNAi in bloodstream-form trypanosomes[11]. In agreement with previous studies, we were unable to generate gene deletion mutants for six protein kinases previously characterised as being essential for promastigotes (CRK1, CRK3, CRK12, MPK4, TOR1 and TOR2)[16,22,30,42,43]. We were, however, able to successfully generate a gene deletion mutant for LmxM.01.0750 and LmxM.32.1710, previously published as essential in *L. major* and *L. infantum* promastigotes respectively[23,44] and failed to isolate a gene deletion mutant for DYRK1, despite this being achieved in *L. infantum*[28]. These may be due to different gene deletion approaches taken allowing time for compensatory mutations or differences between the species. When we consider essentiality within the different protein kinase families, the AGC and CMGC kinases host a high proportion of essential genes. This is perhaps anticipated given that AGC protein kinases mediate diverse and vital cellular functions in eukaryotes and mutations in AGC protein kinases are associated with human disease such as cancer[45]. In addition, CMGC protein kinases include the cyclin-dependent kinases and MAP kinases, which are fundamental to cell cycle progression[46]. In comparison, the NEK protein kinases, which have an expanded family in *Leishmania*[10] are relatively dispensable in *Leishmania* promastigotes with 18/20 gene deletion mutants successfully generated. We noted that protein kinases located in some organelles were more likely to have essential functions than in others, for example, 9/20 (45%) in the nucleus were found to be essential; many of these are CDKs (CRK9, CRK11, CRK12) or other protein kinases likely to be involved in essential nuclear functions such as cell cycle control and transcription (KKT2, KKT3, AUK1, TLK, PKAC1, LmxM.24.0670)[11,47]. In contrast, as the flagellum is not essential for promastigote growth in vitro[15,41], it is not surprising that only 2/13 (15%) protein kinases found in the flagellum are essential. The basal body is another organelle whose correct duplication

and segregation are crucial for cell division[48,49] yet, surprisingly, only 2/23 (9%) of protein kinases localised to the basal body are essential (PLK and LmxM.14.0830).

In order to pool our library of protein kinase gene deletion mutants, and test requirements through life cycle stages, we utilised bar-seq, a powerful technique previously used for genome-wide screens in multiple human cell types, as well as parasites such as *Toxoplasma* and *Plasmodium*[50–52]. This approach has now been applied to a small group of *Leishmania* flagellar proteins[41], ubiquitination pathway enzymes[33,37] and the protein kinases of this study. Whilst the plot of barcode proportions as trajectories across experimental time points provided information on life cycle progression for individual mutants, projection pursuit cluster analyses provided a non-biased way to group the experimental outcomes into clusters of mutants with comparable trends. The largest proportion of mutants were clustered with flat trajectories, indicating no change in fitness and whilst it may be that many of these protein kinases are not important for differentiation or survival, other possible explanations include redundancy of protein kinases regulating the same pathways or their roles in other mechanisms are not related to differentiation, replication and survival. For example, △*clk1* and △*clk2* promastigotes are viable, however, double *CLK1/CLK2* gene deletion mutants could not be isolated, indicating that they might have redundant functions in regulating the kinetochore, as has been described in trypanosomes[47,53]. Five protein kinases have signal peptides and two of the casein kinase family, CK1.4 and CK1.2 are secreted[26,54] and potential virulence factors, however, they are essential in promastigotes and therefore were not present in the pooled libraries. The loss of fitness phenotype for gene deletion mutants of secreted protein kinases may also be masked if they are secreted by other mutants in the pool, still influencing the host-parasite interaction and immune response.

Successful infection of the mouse was the most challenging phenotype assessed for the gene deletion mutants, as the cells required the ability to differentiate from promastigote to amastigote, infect and survive within the macrophage, as well as withstand the host immune response. Using cluster analysis we identified 29 genes required across at least two of the differentiation-specific experimental arms and a further 15 only required for infection of the mouse. It is worth noting that the screen does not distinguish between those mutants that have a significant loss of fitness during differentiation itself and those that differentiate normally and then have a loss of fitness growing as amastigotes. Using these data protein kinases were identified that are involved in extrinsic pathways activated in the parasite by external factors (e.g. from macrophage or sand fly) or intrinsic pathways activated by internal factors (e.g. the state of the cell cycle), independent of external factors. For example, CRK4 is required for infection of the mouse but is not required for axenic amastigote growth, suggesting an extrinsic role. On the other hand, MPK10 clusters as important across all three experimental arms implying an intrinsic requirement, supported by research showing it to have stage-specific activity important in promastigote and amastigote differentiation[13]. Careful consideration is required when interpreting the data from enriched metacyclic promastigote cells, where a cluster of protein kinases was identified that have an increased trajectory, yet has a loss of fitness during infection of macrophages or mice. Whilst some mutants may become bona fide metacyclic promastigotes yet not infect macrophages, it is also possible that enrichment occurred for repressors of differentiation mutants, which have made an early transition towards amastigote morphology and alternative life cycle stages such as haptomonads. An example is *RDK1*, a regulator of amastigote to promastigote differentiation, and a homologue of *T. brucei* RDK1, which has previously been

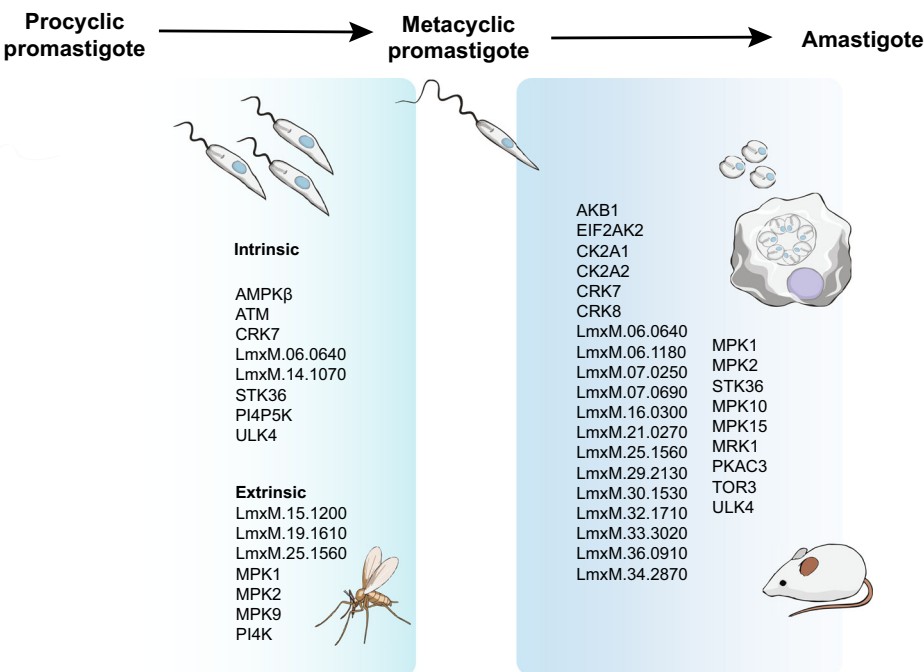

**Fig. 7 Schematic representation of _L. mexicana_ differentiation cycle.** Fifteen protein kinases were identified as required for colonisation of the sand fly. Twenty-nine protein kinases were identified as required for survival as amastigotes in vivo and in vitro.

reported to be involved in differentiation of the bloodstream to procyclic form[11]. Additionally, a mutant that had an impaired proliferation rate as a promastigote would not reach stationary phase by day 7, leading to fewer infective metacyclic forms present in the pooled culture. AMPKα is an example of this, with a consistent loss of fitness throughout the procyclic promastigote stages and metacyclic enrichment, leading to poor representation in macrophage and mouse infections. Phenotyping of selected individual mutants, rather than a pool, will allow more detailed analyses.

Fifteen protein kinases were identified as being required for successful colonisation of the sand fly midgut. As sand flies were fed with procyclic promastigotes the phenotype investigated was the establishment of infection in the insect host; the natural infection of sand flies with amastigotes and the role of protein kinases in the differentiation of amastigote to promastigote was not investigated in this study. Development of _Leishmania_ in the sand fly is far more complex than that observed in in vitro tissue culture and involves sequential differentiation from procyclic promastigote to the nectomonad form and then to the leptomonad form. At day 1 post-infection procyclic forms are numerous within the blood meal surrounded by the peritrophic matrix and by day 3 long nectomonads prevail; these escape from the ruptured peritrophic sac and attach to the midgut epithelium to prevent defecation with the remains of the blood meal[55]. In the next few days, leptomonad promastigotes colonise the anterior part of the midgut and differentiate into non-dividing metacyclic promastigotes in preparation for transmission to a mammalian host. The presence of these highly infective metacyclic parasites, together with the colonisation of the stomodeal valve, is typical for fully developed late-stage infections at day 8[8]. Further investigation will be required to determine the exact roles that these 15 protein kinases have a role in these differentiation, migration and establishment events. Of note, seven of the fifteen protein kinases required for establishment of infection in the sand fly were also required in the amastigote stage, suggesting a stage-independent role in growth and differentiation. LmxM.25.1560, a cytoplasmic CMGC kinase, is notable in being both a likely

pseudokinase and the only _Leishmaniinae_ unique kinase (LUK) where the mutant had a phenotype. Eight protein kinases, including the aPK ATM (LmxM.02.0120), recently associated with a novel DNA damage-induced checkpoint in _Leishmania_[56], were identified as dispensable for full life cycle progression in the mouse but found to be important for development in the sand fly.

The transwell screens identified a number of protein kinase mutants with potential motility defects, including Δmpk3, but surprisingly not Δmkk1; both of these mutants had previously been reported to have reduced motility[15]. We focused on four mutants that had both a short flagellum and severely defective motility, allowing a comparison between the phenotypes in vitro and in the sand fly. Motility is required for survival in the sand fly[41] and indeed we found that Δulk4 and Δstk36 had a loss of fitness for sand fly colonisation, as anticipated. In contrast, Δ29.0600 and Δ02.0570 were non-motile yet were able to colonise the sand fly and did not diminish even after defecation of the blood meal. Further investigation will be required to determine whether the short flagellum provides sufficient binding capacity for attachment to epithelium and colonisation of the midgut. The data on ULK4 and STK36 provide an example of how localisation and life cycle progression data can be used for signalling pathway hypothesis generation. ULK4 and STK36 have very similar infection profiles, localisation and their orthologues were also identified in a flagella attachment proteome in _T. brucei_[57] providing further evidence that they act together in _Leishmania_ and other trypanosomatids.

In summary, we have identified fifteen protein kinases that are required for colonisation of the sand fly, eight of which have an intrinsic role and seven of which are extrinsic protein kinases required for survival in the sand fly only. We also identified 29 protein kinases that are required for successful differentiation from metacyclic promastigote to amastigote and survival in macrophages and the mouse (Fig. 7). These include 5 protein kinases potentially involved in MAP kinase signalling (one of which, MPK2 has a role in the arginine depletion response[18,19]), 2 cyclin-dependent kinases, 2 casein kinase II and 13 protein kinases that have not previously been studied. Thus, a large

databank of information is now available for each of the protein kinases that make up the *Leishmania* kinome, providing an opportunity to decipher signalling pathways and link protein kinases involved in regulating important aspects of the parasite's biology such as replication, differentiation and responses to stress. Perhaps more importantly, these data present an opportunity to characterise and focus experimental effort on those pathways which are divergent to humans, and which potentially can be targeted in new drug and vaccine therapies.

## Methods

**Cell culture**. *Leishmania mexicana* (MNYC/BZ/62/M379) promastigotes were routinely cultivated in HOMEM medium (modified Eagle's medium) supplemented with 10% (v/v) heat-inactivated fetal calf serum (HIFCS – Gibco) and 1% Penicillin/ Streptomycin solution (Sigma-Aldrich) at 25 °C. For differentiation assays, promastigotes were grown in Grace's insect medium (Sigma-Aldrich) with 10% HIFCS (Gibco), 1% Penicillin/Streptomycin solution (Sigma-Aldrich) and 1% BME vitamins (Sigma), pH 5.5 at 25 °C. Parasites were subcultured at $1.5 \times 10^5$ parasites mL$^{-1}$, reaching mid-log phase at 72 h with ~$5 \times 10^6$ parasites mL$^{-1}$, and stationary phase at 168 h with ~$2 \times 10^7$ parasites mL$^{-1}$. Axenic amastigote differentiation and proliferation were achieved by incubating log-phase promastigotes in Schneider's *Drosophila* Medium (Sigma-Aldrich) supplemented with 150 µg mL$^{-1}$ of Hemin and 20% HIFCS at the concentration of $1 \times 10^6$ parasites mL$^{-1}$ for 120 h at 35 °C, 5% CO$_2$. Axenic amastigotes were subcultured by transferring cells to a fresh medium at the same initial concentration. Bone marrow-derived macrophages (BMDM) were differentiated using committed myeloid progenitors isolated from BALB/c mice and incubated with macrophage colony-stimulating factor secreted by L929 cells[58]. Macrophages infected with metacyclic promastigotes isolated by centrifugation over Ficoll 400 (Sigma-Aldrich) gradient[59] and quantified on a hemocytometer. Metacyclic promastigotes were incubated with BMDM (1:1) for 4 h and washed before incubation at 35 °C and 5% CO$_2$ for 12 h and 72 h in RPMI supplemented with 10% HIFCS and 1% Penicillin/Streptomycin solution.

**Generation of the CRISPR–Cas9 mutants and bar-seq screens**. The protein kinase gene deletion library and N- and C-terminal tagged cell lines were generated and validated[33] using http://leishgedit.net/ to design primers in the *L. mexicana* Cas9 T7 line[31]. Primers used to generate sgRNA expression cassettes and donor cassettes, as well as the ones used for diagnostic PCRs, are available in Supplementary Data 7. Primers used to for tagging are available in Supplementary Data 8.

Life cycle phenotyping of gene deletion mutants in vitro and in vivo[33] was carried out as follows: Mid log-phase promastigote mutants ($n = 159$, Pool 1) were grown separately, mixed in equal proportions to give six independent replicates ($4 \times 10^4$ mL$^{-1}$ for each cell line), and then placed in Grace's media supplemented with 20% of FCS, 1% Penicillin/Streptomycin solution and 1% BME vitamins, pH 5.5 at 25 °C. Samples were collected from each of the six replicates at 0, 24, 48, 72 and 168 h. Axenic amastigotes were differentiated from 168 h promastigote cultures and grown for 120 h. Metacyclic promastigotes from six independent 168 h promastigote cultures were purified from Ficoll 400 gradients and samples were collected for time point 0 h. These purified metacyclic promastigotes were used for infection of BMDM at a 1:1 ratio (samples collected at 12 and 72 h) and 12 mice were also infected with 10$^6$ parasites in the left footpad (two animals for each of the six replicate samples, and time points collected at 3 and 6 weeks). DNA was extracted using DNeasy Blood & Tissue (Qiagen) following the manufacturer's instructions for animal cells (promastigotes, axenic amastigotes and macrophage infections) or animal tissue (mice lesions).

For *Lutzomyia longipalpis* infection, promastigote growth data collected in experimental arms 1–3 at 72 h was ranked according to barcode representation and split into Fast, Average and Slow groups accordingly (Supplementary Data 6). Three sub-pools (slow, ave and fast) of protein kinase mutants plus controls were mixed in equal proportions using mid-log procyclic promastigotes ($n = 57$, 57 and 56, respectively). For this experiment (Pool 2), three heterozygous protein kinase mutants were removed and an additional five protein kinase gene deletion mutants that had not been included in the mammalian infection screen were added (Δ03.0210, Δ29.3580, Δ29.3050, Δ20.1340 and Δ36.2630; Supplementary Data 6). The pool final density was counted and 50 mL cultures were set up at $2 \times 10^5$ cells mL$^{-1}$ in HOMEM media. At 72 h, cells were washed three times in PBS and re-suspended in defibrinated heat-inactivated sheep blood at a concentration of $2 \times 10^7$ promastigotes mL$^{-1}$. The *L. longipalpis* colony was maintained at 26 °C and high humidity on 50% sucrose solution and a 12 h light/12 h dark photoperiod, as described previously[60]. The three sub-pools were each fed to a cage of 200 female flies between 3 and 5 d old, through a chick skin membrane. It was assumed that each fly could take ~300 of each mutant in the sub-pool[61]. DNA samples were prepared from the pools prior to sand fly feeding (72 h after pooling, referred to as Time 0) and also from duplicate sets of 25 whole sand flies taken at 1, 5 and 8 d post blood meal. DNA was extracted using a Tissue Lysis Buffer and PCR template preparation kit (Roche).

Preparation of DNA, PCR and sequencing of barcodes was performed as described previously[33]. Amplification of the barcodes was carried out using F primer OL8684

(TCGTCGGCAGCGTCAGATGTGTATAAGAGACAGAgatgtgattacTAATACG ACTCACTATAACTGGAAG) and R primer OL11799 (GTCTCGTGGGCTCGG AGATGTGTATAAGAGACACTCGTTTTCATCCGgcag). Q5 polymerase was used to amplify the barcodes in a 20 ul reaction containing 100 ng of DNA, with 28 cycles of amplification, 60 °C annealing temperature and 10 s extension time. Nextera adapters were then added to the resulting DNA before being analysed on an Illumina HiSeq 3000. A Python script was used to search each Illumina read for the 12 bp sequence preceding each barcode (Supplementary Software 1). Total counts for each unique barcode are provided in Supplementary Data 6.

**Animal work**. All experiments were conducted according to the Animals (Scientific Procedures) Act of 1986, United Kingdom, and had approval from the University of York Animal Welfare and Ethical Review Body (AWERB) committee. To assess virulence of the *Leishmania mexicana* Cas9 T7 progenitor line, Δ*rdk1* mutant and an add-back Δ*rdk1::RDK1* groups of 5 female BALB/c mice (4–6 weeks) were infected subcutaneously at the left footpad with $1 \times 10^6$ purified metacyclic promastigotes. Animals were culled after 6 weeks and the parasite burden determined with a limiting dilution assay essentially as described previously[62]. Footpads were digested with 4 mg mL$^{-1}$ collagenase D for 2 h at 37 °C before dissociation through 70-µm cell strainers (BD Biosciences). Homogenates were re-suspended in HOMEM supplemented with 20% FCS and serial dilutions (2-fold) performed in 96-well clear flat-bottom plates. Dilutions were performed in duplicate, distributed in at least 4 plates and incubated for 7–10 days at 25 °C. Wells were analysed for the presence of parasites, and the number of parasites calculated by multiplying the dilution factors. Results are shown as total parasites per tissue. Two-tailed unpaired *t*-test was performed as statistical analysis.

**Bar-seq statistical analyses**. For the purposes of clustering, we characterised mutant phenotypes in terms of the changes over time of the logged-proportion of each mutant subpopulation relative to the total population. For each mutant, these changes constitute a time series to which we apply a differencing operator that removes the time series' average heights and slopes so that attention is instead focused on their curvature. The differenced times series are then treated as high-dimensional Euclidean vectors. These vectors are clustered using a Projection Pursuit Clustering algorithm, adapted from ref. [39] and coded up in R (R: The R Project for Statistical Computing, n.d.). The algorithm, for which pseudo-code is provided in Supplementary Equation 1, involves iteratively projecting and re-clustering the vectors. This method is used to find a plottable two-dimensional space in which the phenotype vectors are maximally separated. Once projected into this space, proposed clustering solutions for the vectors can be scrutinised and checked against prior knowledge of genetic characteristics. Such scrutinising led to the decision to choose a 6-cluster solution from the algorithm, whose output is illustrated in Supplementary Data 5.

**Whole-genome sequencing of protein kinase mutants**. Sequence reads generated from an Illumina HiSeq 3000 were trimmed with Cutadapt version 2.5[63] and mapped to the *L. mexicana* Cas9 T7 genome assembly[64] and resistance cassettes with BWA-MEM algorithm as part of bwa version 0.7.17[65]. Mosdepth 0.2.6[66] was used to calculate the median coverage of each chromosome and scaffold in the assembly, as well as the mean coverage of each gene of interest. The ratio of gene coverage to chromosome coverage was used to assess gene deletion.

**DNA cloning**. Expression of protein kinases either for facilitated gene deletion mutant or addback generation was achieved by using the pNUS system[67] modified for expression with a C-terminal 6xMYC fusion. Gibson Assembly (NEBuilder® HiFi DNA Assembly) was used. The pNUS backbone plasmid was digested using AvrII and gel extracted (Qiagen). The required CDS was amplified from *L. mexicana* genomic DNA using primers designed on NEBuilder assembly online tool, with a 5′ 20 nt overlap with the pNUS backbone. Fragments were purified (Qiagen) and Gibson Assembly was performed as suggested by the manufacturer and transformed into *E. coli* DH5α chemo competent.

**Live cell imaging**. Endogenous tagged cell lines in log-phase were centrifuged at $1000 \times g$ for 10 min, washed in phosphate saline buffer pH 7.2 (PBS) and re-suspended in 10 µg mL$^{-1}$ of Hoechst 33342 (ThermoScientific) before incubation for 15 min at room temperature protected from light. To remove Hoescht 33342 the cells were washed with PBS, re-suspended and immobilised in 50 µL of ice-chilled CyGEL$^{TM}$ (Biostatus). Samples were transferred to slides, covered with a coverslip and immediately imaged using a Zeiss LSM880 confocal microscope, with 488 and 405 nm lasers, at room temperature. Images were processed using Zen Black (Zeiss).

**Western blotting**. Cells were re-suspended in NuPAGE sample buffer supplemented with 50 mM of dithiothreitol (DTT) and loaded on 4–12% NuPAGE Bis-Tris Protein Gels and electrophoresed at 180 V for 1 h. Gels were transferred to Nitrocellulose membranes using wet transfer system XCell II Blot Module (Invitrogen) at a constant amperage of 350 mA for 2 h. Membranes were blocked with 5% skimmed milk in Tris Buffered saline-Tween for 1 h before incubation with

primary antibody anti-myc tag monoclonal clone 4A6 (Millipore) diluted to a final concentration of 200 ng mL$^{-1}$ in 3% skimmed milk in TBST for 1 h at room temperature. After washing membranes proceeded to incubation with secondary antibody anti-mouse IgG (H + L) conjugated to HRP (Molecular Probes) at 300 ng mL$^{-1}$ in 3% skimmed milk in TBST for 1 h at room temperature. Membranes were washed, incubated with Clarity Max substrate (BioRad) and developed in ChemiDoc MP (BioRad).

**Motility screen and assays**. Six replicates were prepared for Pool 1. Cultures were grown in HOMEM media for 12 h in 20 mL culture. A 2 mL culture was added to the bottom of each well of an 8-μm transwell plate (Costar #3428 24 mm Diameter insert, 8.0 μm Pore size 6 well plate) and additional media was added to the top of each well before incubating at 25 °C for 4 h to allow motile cells to cross the membrane and equilibrate on both sides. The culture was collected from the top and bottom of each well. Cells were pelleted, DNA extracted and bar-seq analyses completed. Statistical analyses included two-tailed unpaired t-test and discovery using the two-stage linear step-up procedure of Benjamini, Krieger and Yekutieli, with $Q = 50\%$. Each row was analysed individually, without assuming a consistent SD. Outputs with a p value < 0.005 were chosen for further phenotypic characterisation. This experiment was repeated with Pool 2, using 2 replicate wells.

Motility mutants were grown individually to a mid-log culture. Cells were fixed in a final concentration of 1% paraformaldehyde (Alfa Aesar 16% W/V #43368) for 15 min. Fixed cells were washed in PBS by pelleting at $1000 \times g$ for 1 min. The remaining pellet was gently re-suspended in 20 μL PBS. 10 μL of suspension was added to a glass slide (Thermo Superfrost plus #J1800AMNZ) with 5 μL of mounting solution and coverslip. Cells were imaged on an inverted Zeiss AxioObserver with a ×63 lens. Multiple field of view images were taken for each line. Cell morphology measurements were made using Zen-2.6 (blue-edition). For consistency, cell width was always measured across the nucleus, while flagellum length was measured from the tip to the edge of the cell body. Statistical analyses included two-tailed unpaired parametric t-test with Welch's correction. Each row was analysed individually, without assuming a consistent SD.

For tracking, cells were loaded into glass sample chambers measuring ~5 mm × 20 mm × 0.5 mm constructed from glass slides and UV-curing glue. The samples were imaged under darkfield illumination on a Nikon Eclipse E600 upright microscope using a ×10 magnification objective lens with a numerical aperture of 0.3. Five video microscopy sequences were acquired from different areas of the sample chamber for each mutant using a CMOS camera (Mikrotron MC-1362). Movies were acquired at 50 Hz for 40 s, at a resolution of 1024 × 1024 pixels, corresponding to a field of view of ~1.4 mm × 1.4 mm within which cells were tracked. Minimum images were constructed, composed of the lowest values at each pixel in each video sequence. A rolling ball filter with a radius of 30 pixels was applied to each minimum image before using them to correct the video sequences and remove static background features. The corrected movies were then thresholded to highlight cells moving by either swimming or Brownian motion. The cell coordinates in each frame were extracted, and these coordinates were assembled into cell tracks using custom-written software routines[68]. To smooth short-time fluctuations associated with camera noise, the cell tracks were fitted using piecewise cubic splines. This allowed reliably extracted instantaneous cell velocities. This technique also allows the separation of motile and non-motile cells based on their mean-squared displacements. The spline-smoothed trajectory of a non-motile particle has a mean-squared displacement that scales with time, whereas a swimming cell's mean-squared displacement scales with time squared. By setting a threshold for the slope of each cell's mean-squared displacement at short times, and a threshold for the total mean-squared displacement after 2 s, motile and non-motile particles were effectively separated and the motile fraction estimated. The instantaneous speed of each swimming cell was averaged over the course of the cell track, and distribution of average swimming speeds made.

**Reporting summary**. Further information on research design is available in the Nature Research Reporting Summary linked to this article.

## Data availability

Whole-genome sequencing data for *L. mexicana* Cas9T7 and gene deletion mutants, as well as the bar-seq data are available from the European Nucleotide Archive under study accession PRJEB40373. All other data generated or analysed during this study are included in this published article (and its Supplementary Information files). Source data are provided with this paper.

## Code availability

All code used during this study is included in this published article (and its Supplementary Information files).

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

## Acknowledgements

This work was supported by the Wellcome Trust (200807) to J.C.M. and the EPSRC (EP/N014731/1) to L.G.W. The research has, in part, received funding from the Research Council United Kingdom Grand Challenges Research Funder under grant agreement 'A Global Network for Neglected Tropical Diseases' grant number MR/P027989/1. J.S., B.V. and P.V. were supported by ERD Funds, project CePaViP (CZ.02.1.01/0.0/0.0/16_019/0000759). This project has also received resources funded by the European Union's Horizon 2020 research and innovation programme under grant agreement No 731060 (Infravec2). We thank Peter Ashton, Sally James, Lesley Gilbert, Karen Hogg, Graeme Park, Karen Hodgkinson and Sarah Forrester for technical advice and support within the Bioscience Technology Facility, University of York.

## Author contributions

J.C.M. conceived the study. N.B., C.C.M.C.-P., R.N., E.V.C.A.F., V.G. and J.B.T.C made the kinome libraries. N.B., C.C.M.C-P. and R.N. A.J. carried out the bar-seq screens. B.P. carried out statistical analysis. N.B., C.C.M.C-P., B.P., K.N. and J.W.P analysed data. J.S., B.V. and P.V did sand fly experiments. P.B.W and L.G.W did motility analysis. N.B., C.C.M.C-P., R.N., V.G. and A.M. did phenotype analysis. N.B., C.C.M.C-P. and J.C.M prepared and wrote the manuscript. All authors reviewed, edited and approved the paper. J.C.M. obtained funding.

## Competing interests

The authors declare no competing interests.
