## [Peer Review File · Nature Communications]

Reviewer comments, first round:

Reviewer #1 (Remarks to the Author):

This is a well-written manuscript reporting on a kinome-wide functional study in *Leishmania mexicana*. The methodology is sound and relies on elegant and cutting-edge approaches, the data are solid and interpreted with caution. The study provides a comprehensive map of the role of all members of the parasite's kinome (i) in the transition between developmental stages of the parasite during its life cycle, (ii) during infection of the insect vector and the vertebrate host, and (iii) in parasite motility. Overall, this fills in a notable knowledge gap in the biology of *Leishmania* and generates a vast number of testable hypotheses with respect to cellular function of the parasite's kinases, as well as prioritising kinases as potential targets for chemotherapeutic intervention. As such, the paper is likely to have considerable impact. The high level of congruence between on the one hand data obtained in this study with respect to essentiality at the promastigote stage, and on the other hand published data on individual genes (line 321), strongly supports the validity of the functional screen presented here. It is not surprising that there were discrepancies, rightly explained (line 326) by the differences in approaches --this has been observed in other systems (e.g. with respect to *Plasmodium* essentiality of some kinases at various developmental stages). The study also allowed a refinement of the genome in some instances (e.g. CLK1 locus, lines 162-167).

This reviewer has little doubt that this will prove a seminal paper in the field, and has no major issue with this overall excellent study. However, the author might consider the following points, which in the view of this reviewer would improve the manuscript:

1. *Leishmania mexicana* kinome: It is not clear how the kinome was assembled. Lines 113-115 mention that "ePKs, PIKKs and aPKs sequences were retrieved using *L. major*, *T. brucei* and *T. cruzi* gene IDs". How was that achieved? Is every gene in the *L. mexicana* database assigned to orthologous groups with genes for the three other species? A targeted de novo search of the *L. mexicana* predicted proteome, e.g. HMM using PK signatures, might identify additional, *L. mexicana*-specific PK-encoding genes.
2. Fig. 2c: The heatmap does indeed, as stated (line 136), "confirm loss of reads over the CDS of interest" (i.e. the diagonal of light squares) in the 18 lines whose whole genome was sequenced. It is intriguing to this reviewer that some kinases seem to be upregulated in some of the deletion mutants. For example, kinase 20.1340 seems to be elevated (in terms of coverage index) in $\Delta 07.0900$, $\Delta 29.0600$ and $\Delta mkk1::MKK1$. There are other examples. Does this reflect a possible complementation effect / epistatic interaction between 20.1340 and these three kinases? There are examples in the literature where deletion of one kinase is possible, but another kinase of the same group is overexpressed in the mutant (e.g. *Plasmodium* MAPKs, PMID: 17651389). It would be interesting to determine if this is the case for *ckl1* and *clk2* (for example) -- a qRT-PCR would suffice... This is not a requirement for this paper, but would be of interest (maybe in a subsequent study of *Lm* CLKs), and deserves a comment in the present paper. It would substantiate the statement (line 327) that compensatory mechanisms may occur to allow for deletion of specific kinases.
3. The study allowed a refinement of the genome in some instances (e.g. CLK1 locus, lines 162-167). Will this be used to update the genomic database?
4. Localisation studies: The images are mostly clear (and beautiful). The transfer from the cytoplasm to the nucleus of some kinases (e.g. MPK6) is reminiscent of nuclear translocation of mammalian MAPKs upon activation. Does the nuclear re-localisation of MPK6 in S-phase correlate with its biological function (if known)? If so, this (and other instances of re-localisation) would definitely deserve a short discussion.
5. Survival in the sandfly: The rationale for splitting the mutant set into three subpools according to growth rate at the promastigote stage (line 265) makes sense, as this stage is replicative in the sand fly --this reviewer assumes this strategy is designed to minimise competition from fast-

growing parasites that would skew the data (if this is right, this may deserve a brief explanation after line 265). However, first, the growth rate may be different in the fly's midgut than it is under laboratory culture conditions. Second, the natural infective form (from the vertebrate host to the sandfly) is the amastigote. It appears from the methods description (lines 463-479) that flies were fed artificially with blood spiked with cultured promastigotes. Would this strategy miss kinases required for natural infection from the amastigote stage? A comment to this effect would be useful.

6. Line 394 (and other discussions of pseudokinases): it may be useful to consider predicted pseudokinases that actually turned out to possess kinase activity despite lacking "critical" residues (e.g. PMID: 24567368; PMID: 20421461, entitled "yet another active pseudokinase"...). Are some of these substitutions present in some of the Leishmania pseudokinases? Could some of these proteins be active?

Editorial points

Abstract, line 35. It is important to state in the abstract that the identification of genes as dispensable or refractory to deletion is based on the data from gene targeting in promastigotes.

The last sentence of the Abstract ("...allows targeted investigation of organelle-associated intrinsic and extrinsic signalling pathways...") is somewhat obscure; it becomes clearer after the manuscript is read, but this part of the Abstract should be made clearer/more accessible. See below an additional comment about "extrinsic".

Line 54: "... undergo cell cycle arrest in response to autocrine signals in their host". This slightly confusing. Which autocrine signals? Secreted by the parasite or by the host? Please clarify.

Line 68: "...reach the cardia of the sand fly to be transmitted...." Not clear to the non-specialist reader. Briefly define "cardia", and how this facilitate transmission.

Line 70: Before the sentence "Leishmania has adapted...", a brief statement saying that after entering the human host the parasite is phagocytosed by macrophages would be helpful to the non-specialist reader.

Line 75: Phosphorylation-mediated (hyphen)

Line 84: It would be of interest to mention the absence of TyrKs in other unicellular parasites (e.g. Plasmodium, PMID: 15479470; Trypanosoma spp, PMID: 16164760).

Line 115, line 141, line 139 and abstract line 36: Line 115 mentions "195 kinases investigated". Line 141 mentions success to generate deletion mutants for 162 kinases, suggesting there are $195-162=33$ that are refractory to deletion kinases; but the line 139 and the abstract mention 44 kinases refractory to deletion. Please clarify.

Line 133: could Southern blot or qPCR data be generated to substantiate the possibility of chromosome/gene duplication? This is not a requirement, since it is been documented in other Leishmania studies, but if data are available it would be good to mention it.

Line 186: please explain what the non-kinase control (Δmca) is.

Line 208: A very brief explanation of the principle of the "projection pursuit method" would be helpful to the non-specialist reader.

Lines 184-237: this section is very long, dense and difficult to follow. I would suggest breaking it into section, e.g. by gene families. (however, Fig 5 nicely summarises these data).

Lines 370 and 372: Please clarify "extrinsic" –does this mean signalling pathways of the host cell? If so, the authors might want to cite PMID 32782246, which describe a large extent of host cell signalling in another parasitic context.

Line 390: for the non-specialist reader, please define "leptomonad", "heptomonad", "retro-leptomonad".

Line 406: "...associate with others in the pool." Do the authors propose that parasites physically attach to each other, and that one that has no motility can, as it were, hitch a ride? Are there any observations that may substantiate this intriguing possibility?

Report written by Christian Doerig

Reviewer #2 (Remarks to the Author):

This manuscript from Baker et al reports the results from an genome-scale interrogation of the all 206 genes encoding protein kinases in the *Leishmania mexicana* genome. As such, it represents enormous experimental effort, and the results reported will provide a tremendous resources for other researchers in the field. The manuscript is well-written (with only a few examples where the language could be improved) and the data analysis appears solid. All six Figures are well-chosen to represent examples of the much greater amount of data available as supplementary information. I recommend publication without revision.

Reviewer #3 (Remarks to the Author):

In this study, the authors systematically deleted all individual kinase genes from the *L. mexicana* genome and tested their fitness in vitro, in macrophages, in a mouse infection model and in sand flies. In addition kinases were tagged with a fluorescent protein to record their cellular localisations.

This is an important milestone in genomic research in *Leishmania*. It shows that 79% of kinase genes are non-essential in culture promastigotes, and identifies 29 kinases important for amastigote survival and 15 important for survival in the sand fly vector. This is the first global overview of the relative importance of different kinases in the parasite's life cycle and is therefore of significant value as a data resource that will allow for further targeted investigations both on the biology and on potential drug targets.

There are a few areas where a clarification would enhance the paper:

1. Gene KOs

1.1. Technically, this work is sound and generally well-documented. Explanations would be helpful on the following points:

- An easier-to-follow breakdown of the number of kinases studied. The total from Fig 1c appears to be: $162+44=206$. In the text (lines 115-120) this matches the $195 \text{ ePK} + 11 \text{ aPK} = 206$. What then does it mean "We also identified" 27 found only in *Leishmania* + 17 LUKs? And in Fig 1b, what do the blue circles actually mean and why do they add up to more than the total number of kinases analysed ($174 + 27 + 17 + 1 = 219$)?
- Fig 2b, what do the colons mean in $\Delta\text{pkac1}::\text{PKAC1}$? Is this to indicate that only one allele was deleted? It reads like a fusion.
- Supplementary Data 1, PCR validation, please indicate sizes for the DNA ladder as reference and state the expected band sizes (here or in the primer table).
- For the 44 kinases refractory to deletion, it would be important to state whether obvious technical reasons for failure can be excluded (are any in arrays, was the targeting sequence checked against the nanopore sequenced genome, etc?). Examples are given for three genes (lines 159-) but it is not clear whether this was systematically checked.

2. Gene Tagging

The localisation of 199 tagged kinases is in itself a valuable dataset. There is however relatively little analysis of that data, only a short paragraph in the Results.

2.1. Cell cycle dependent localisations for some kinases, which are mentioned in the text, should be supported by the inclusion of the relevant images.

2.2. There are issues with the annotated localisations in Supplementary Data 2:

The localisations don't always match what is visible in the pictures and they also don't match the terms shown in Figure 3b. A non-exhaustive list of examples:

A6-A9, nucleoplasmic localisation not evident

A11, AMPGK, CBPK1, LmxM.21.0853, flagellar cytoplasm fluorescence not visible

C2, only nucleoplasm seems fluorescent

CK1.4, ATM, no cytoplasmic fluorescence visible

LmxM.33.3020, nucleoplasm not fluorescent

MKK4, LUK6, no cytoplasmic and flagellar fluorescence visible

The distinctions between flagellar pocket / endocytic not always clear, also basal body/kinetoplast.

The criteria for defining a localisation as "endomembrane" are not clear. How was "flagellar cytoplasm" defined and how was it distinguished from axoneme or PFR?

I would recommend checking these again, within the set and also against the "landmarks" reference paper. The simplest approach may be just to use the terms shown in Figure 3b consistently.

2.3. Cyan (DNA) and green fluorescence is hard to distinguish in the overlay, a combination for example of purple / green would be clearer (like in Fig 3c, which shows very nice examples of the different localisations).

2.4. Most cells show a very elongated morphology. Is it certain these images were taken from actively dividing mid-log cultures, not stationary cultures? This may affect observed localisation.

3. Phenotyping – fitness in vitro and in vivo

The analysis focuses on survival in different life cycle stages, in culture, in cells, in mice and in sand flies. The design is elegant, tracking populations from promastigotes continuously through to different end-points with relevance to the actual life cycle.

3.1. The following clarifications would be helpful:

- Why were only 154 gene deletions out of 162 included in pooled screens?
- What is the significance of the Δmca control? What does its trajectory look like and did it correspond to expectations?
- Were barcoded control lines included (as for sand fly infections) and what does their trajectories look like?

3.2. For the fly infections, mutants were separated into three pools based on growth rate. Why, and was this also done for the other pooled screens? It is striking that there doesn't seem to be a "cluster" of mutants with reduced fitness in promastigotes but average or enhanced fitness in amastigotes. Could it be an intrinsic limitation of the assay that such mutants were systematically missed?

4. The role of kinases in specific processes

Differentiation

4.1. In the title, abstract & introduction, there is a strong focus on differentiation, yet the paper contains only limited data that speaks directly to differentiation. (Selection of metacyclics and the detailed examination of $\Delta rdk1$, below). Under the Heading "Protein kinases important for differentiation from promastigotes to amastigotes", the paper reports 29 kinases important for amastigotes. This is clearly a very important finding, not least for drug discovery campaigns. It does not tell whether differentiation or other processes were affected. The title of the paper ("regulators of differentiation") doesn't accurately reflect the key findings.

4.2. For $\Delta rdk1$, the data provide strong evidence supporting a critical role of *rdk1* in amastigotes. But do these mutants really have accelerated differentiation from amastigotes to promastigotes? The HASPB blot which is presented as evidence supports this to some extent (although the unequal loading of the lanes raise the question how reproducible this was). I would be interested to know whether this acceleration was evident in an accelerated morphological change, which provides independent and easily measured markers (elongation of cell body and outgrowth of the flagellum).

A growth curve for axenic amastigotes (KO, WT and add-back) would show whether slower replication of $\Delta rdk1$ amastigotes could explain the lower parasite numbers in macrophages and mice, rather than a differentiation defect.

Line 148, "identifies RDK1 as a Leishmania repressor of differentiation" – it is unclear what differentiation is being repressed (presumably amastigote to promastigote) and how that links to the observed defects in amastigotes.

4.3 Motility. An innovative trans-well assay was used to identify motility mutants from mutant pools. Further clarification on the details and validation would help to assess the power and limitations of this assay:

- Were all mutants included and is the full bar seq data available? In the Supplementary Table I can only see the "significant motility outputs" but not the full dataset.
- What controls were done to validate this assay? Were any known paralysed, slow and fast swimming mutants tested to assess the discriminatory power of the assay?
- Mutants with a very short or absent flagellum are easily recognised just by microscopic examination of the culture. Was this recorded for the mutants, which were generated as individual cell lines, and how did the results match up with the trans-well assay?
- Wiese's group previously reported that deletion of MPK3 and MKK1 resulted in very short flagella (refs 12 and 13) and reduced motility. Were these mutants present in this screen and could the authors comment on why they might have been missed in the trans well assay? Similar for PKA which was reported to have reduced motility (ref 42).

5. Data accessibility

5.1. The supplementary Excel files contain very valuable data but should be properly labelled, with legends containing the minimal information to allow easy understanding of content, specifically Table legends stating what each Tab of Tables 1 and 2 contain, and supplementary data should be referenced in the text (e.g. morphology measurements for the identification of motility mutants).

5.2. While it is stated that "additional images are available [...] upon request", it would be preferable to supply these additional images through a suitable repository.

6. Minor comments & typos

6.1 S5 Figure, the μ in the x Axis label doesn't show up properly.

6.2. 178, flagella (not flagellar)

6.3. line 348, do the authors mean: ...leading to fewer infective metacyclic forms (rather than "less infective metacyclic forms")?

6.4. Fig 6a Y-axis label, should presumably be the other way round (Flagella : Cell body ratio, as in S6 a)

6.5. Kinase orthology: Please specify how orthology was established and specify more precisely what "Conserved across everything (123)" means (conserved across surveyed kinetoplastids?).

6.6. Supplementary Figure 4 – the legend refers to mutants highlighted in red but the image is in black and white.

We are grateful to the reviewers for their comments and we have made the following changes (highlighted in yellow in the manuscript).

Reviewer #1 (Remarks to the Author):

This reviewer has little doubt that this will prove a seminal paper in the field, and has no major issue with this overall excellent study. However, the author might consider the following points, which in the view of this reviewer would improve the manuscript:

1. *Leishmania mexicana* kinome: It is not clear how the kinome was assembled. Lines 113-115 mention that “ePKs, PIKKs and aPKs sequences were retrieved using *L. major*, *T. brucei* and *T. cruzi* gene IDs”. How was that achieved? Is every gene in the *L. mexicana* database assigned to orthologous groups with genes for the three other species? A targeted de novo search of the *L. mexicana* predicted proteome, e.g. HMM using PK signatures, might identify additional, *L. mexicana*-specific PK-encoding genes.

Response: The L. mexicana kinome was based on searching the L. mexicana genome (Rogers et al., 2012) in TritypDB for orthologous protein kinases previously identified in L. major (Parsons et al., 2005) and cross checked with L. infantum (Borba et al, 2019). Comparative genomics has revealed that there are no species specific Leishmania protein kinases. We also cross-checked that all L. mexicana proteins with a pfam protein kinase domain are present in the dataset. We have modified the results section to make this clearer.

2. Fig. 2c: The heatmap does indeed, as stated (line 136), “confirm loss of reads over the CDS of interest” (i.e. the diagonal of light squares) in the 18 lines whose whole genome was sequenced. It is intriguing to this reviewer that some kinases seem to be upregulated in some of the deletion mutants. For example, kinase 20.1340 seems to be elevated (in terms of coverage index) in $\Delta 07.0900$, $\Delta 29.0600$ and $\Delta mkk1::MKK1$. There are other examples. Does this reflect a possible complementation effect / epistatic interaction between 20.1340 and these three kinases? There are examples in the literature where deletion of one kinase is possible, but another kinase of the same group is overexpressed in the mutant (e.g. *Plasmodium* MAPKs, PMID: 17651389). It would be interesting to determine if this is the case for *ckl1* and *clk2* (for example) -- a qRT-PCR would suffice... This is not a requirement for this paper, but would be of interest (maybe in a subsequent study of *Lm* CLKs), and deserves a comment in the present paper. It would substantiate the statement (line 327) that compensatory mechanisms may occur to allow for deletion of specific kinases.

Response: We also find this intriguing and have plans to follow up in the future. As pointed out by the reviewer, copy number changes could influence expression of one protein kinase as a way to compensate for loss of another. We are also interested to know if specific SNPs might arise in compensation too. We have included the cas9T7 line in an updated figure 2c. We have also submitted the NGS sequence data to the ENA database. We have added an additional comment in the results to reflect these possibilities.

3. The study allowed a refinement of the genome in some instances (e.g. CLK1 locus, lines 162-167). Will this be used to update the genomic database?

Response: The L. mexicana nanopore genome on which the refinement of the CLK1 and CLK2 locus was based, is available via LeishGEdit. The nanopore genome has also been submitted to TritypDB and should be available shortly.

4. Localisation studies: The images are mostly clear (and beautiful). The transfer from the cytoplasm to the nucleus of some kinases (e.g. MPK6) is reminiscent of nuclear translocation of mammalian MAPKs upon activation. Does the nuclear re-localisation of MPK6 in S-phase correlate with its biological function (if known)? If so, this (and other instances of re-localisation) would definitely deserve a short discussion.

Response: MPK6 has not been studied in Leishmania and we are not aware of any other instances described in the parasite.

5. Survival in the sandfly: The rationale for splitting the mutant set into three subpools according to growth rate at the promastigote stage (line 265) makes sense, as this stage is replicative in the sand fly --this reviewer assumes this strategy is designed to minimise competition from fast-growing parasites that would skew the data (if this is right, this may deserve a brief explanation after line 265). However, first, the growth rate may be different in the fly's midgut than it is under laboratory culture conditions. Second, the natural infective form (from the vertebrate host to the sandfly) is the amastigote. It appears from the methods description (lines 463-479) that flies were fed artificially with blood spiked with cultured promastigotes. Would this strategy miss kinases required for natural infection from the amastigote stage? A comment to this effect would be useful.

Response: We split the mutant set into three promastigote pools for practical reasons, as much as for technical reasons, as the pools were shipped from York to Prague and then used to infect sand flies immediately. Promastigote-initiated infections are a standard for all experiments done with Leishmania mutants. In addition, Sadlova et al. (2017) studied to what extent promastigote-initiated experimental infections differ from those initiated with amastigotes. Both groups developed heavy late-stage infections with the same localization, uniform representation of infective metacyclic forms and equal efficiency of transmission. Whilst in our present study some difference between promastigote and amastigote derived infections might be expected on day 1, the final outcome of Leishmania development on day 8 would be the same.

6. Line 394 (and other discussions of pseudokinases): it may be useful to consider predicted pseudokinases that actually turned out to possess kinase activity despite lacking "critical" residues (e.g. PMID: 24567368; PMID: 20421461, entitled "yet another active pseudokinase"...). Are some of these substitutions present in some of the Leishmania pseudokinases? Could some of these proteins be active?

Response: 11 of the pseudokinases lack all three of the residues thought to be required for activity and which we classify as likely to be inactive kinases. A further 40 are missing one or more residues and we provide a qualification in the text to say that these are putatively active. It will certainly be interesting to follow up those kinases and investigate further

Editorial points

Abstract, line 35. It is important to state in the abstract that the identification of genes as dispensable or refractory to deletion is based on the data from gene targeting in promastigotes.

Response: We have made this clear in the abstract

The last sentence of the Abstract (“...allows targeted investigation of organelle-associated intrinsic and extrinsic signalling pathways...”) is somewhat obscure; it becomes clearer after the manuscript is read, but this part of the Abstract should be made clearer/more accessible. See below an additional comment about “extrinsic”.

Response: We have removed intrinsic and extrinsic from the abstract as it is difficult to explain their meaning in the context of this study within the 150 word limit.

Line 54: “... undergo cell cycle arrest in response to autocrine signals in their host”. This slightly confusing. Which autocrine signals? Secreted by the parasite or by the host? Please clarify.

Response: This refers to parasite secreted signals – we have edited the sentence to make this clear.

Line 68: “...reach the cardia of the sand fly to be transmitted...” Not clear to the non-specialist reader. Briefly define “cardia”, and how this facilitate transmission.

Response: We describe this more and have included an explanation about its role in transmission.

Line 70: Before the sentence “Leishmania has adapted...”, a brief statement saying that after entering the human host the parasite is phagocytosed by macrophages would be helpful to the non-specialist reader.

Response: we have adjusted the sentence

Line 75: Phosphorylation-mediated (hyphen)

Response: corrected

Line 84: It would be of interest to mention the absence of TyrKs in other unicellular parasites (e.g. Plasmodium, PMID: 15479470; Trypanosoma spp, PMID: 16164760).

Response: We have made reference to Plasmodium and Trypanosomes

Line 115, line 141, line 139 and abstract line 36: Line 115 mentions “195 kinases investigated”. Line 141 mentions success to generate deletion mutants for 162 kinases, suggesting there are $195-162=33$ that are refractory to deletion kinases; but the line 139 and the abstract mention 44 kinases refractory to deletion. Please clarify.

Response: We have clarified the numbers.

Line 133: could Southern blot or qPCR data be generated to substantiate the possibility of chromosome/gene duplication? This is not a requirement, since it is been documented in other Leishmania studies, but if data are available it would be good to mention it.

Response: We do not have Southern blot or qPCR data to substantiate chromosome/gene duplication data. We think the NGS data is substantial in its own right (see Fig 2c).

Line 186: please explain what the non-kinase control (Δmca) is.

Response: mca is metacaspase. We have added additional explanation as to why this mutant was added.

Line 208: A very brief explanation of the principle of the “projection pursuit method” would be helpful to the non-specialist reader.

Response: We have added a brief explanation.

Lines 184-237: this section is very long, dense and difficult to follow. I would suggest breaking it into section, e.g. by gene families. (however, Fig 5 nicely summarises these data).

Response: We have split into sections as suggested

Lines 370 and 372: Please clarify “extrinsic” –does this mean signalling pathways of the host cell? If so, the authors might want to cite PMID 32782246, which describe a large extent of host cell signalling in another parasitic context.

Response: In this instance Extrinsic pathways are activated in the parasite by external factors (eg macropahge or sand fly), whereas intrinsic pathways are activated by internal factors (eg state of the cell cycle), independent of external factors. In this case we distinguish parasites in culture vs parasites in the mammalian or sandfly host. We have explained this more clearly in the text.

Line 390: for the non-specialist reader, please define “leptomonad”, “heptomonad”, “retro-leptomonad”.

Response: We have added extra text in the discussion to explain that these are different forms of the parasite found in the sand fly and their relevance to the 8 day infection.

Line 406: “...associate with others in the pool.” Do the authors propose that parasites physically attach to each other, and that one that has no motility can, as it were, hitch a ride? Are there any observations that may substantiate this intriguing possibility?

Response: That is the hypothesis – as we have no evidence to support this possibility, we have removed in from the discussion.

Reviewer #2 (Remarks to the Author):

This manuscript from Baker et al reports the results from an genome-scale interrogation of the all 206 genes encoding protein kinases in the Leishmania mexicana genome. As such, it represents enormous experimental effort, and the results reported will provide a tremendous resources for other researchers in the field. The manuscript is well-written (with only a few examples where the language could be improved) and the data analysis appears solid. All six Figures are well-chosen to represent examples of the much greater amount of data available as supplementary information. I recommend publication without revision.

Reviewer #3 (Remarks to the Author):

There are a few areas where a clarification would enhance the paper:

1. Gene KOs

1.1. Technically, this work is sound and generally well-documented. Explanations would be helpful on the following points:

- An easier-to-follow breakdown of the number of kinases studied. The total from Fig 1c appears to be: $162+44=206$. In the text (lines 115-120) this matches the $195 \text{ ePK} + 11 \text{ aPK} = 206$. What then does it mean “We also identified” 27 found only in Leishmania + 17 LUKs? And in Fig 1b, what do the blue circles actually mean and why do they add up to more than the total number of kinases analysed ($174 + 27 + 17 + 1 = 219$)?

Response: We have rewritten the text to make the number of protein kinases clearer. The numbers in the blue circles show the number of protein kinases that are orthologous between the species (ie 174 are found in trypanosomes and leishmania, whereas only 1 is found in leishmania and endotrypanum but not the other species).

- Fig 2b, what do the colons mean in $\Delta pkac1::PKAC1$? Is this to indicate that only one allele was deleted? It reads like a fusion.

Response: This indicates that two alleles have been deleted ($\Delta pkac1$), but that one chromosomal allele remains $::PKAC1$. We have defined this in the text.

- Supplementary Data 1, PCR validation, please indicate sizes for the DNA ladder as reference and state the expected band sizes (here or in the primer table).

Response: we have included the sizes for the DNA ladder in Supplemental data 1 and included the expected band sizes in Supplemental table 1 (gene deletion primers tab).

- For the 44 kinases refractory to deletion, it would be important to state whether obvious technical reasons for failure can be excluded (are any in arrays, was the targeting sequence checked against the nanopore sequenced genome, etc?). Examples are given for three genes (lines 159-) but it is not clear whether this was systematically checked.

Response: This was indeed checked systematically. All targeting sequences were derived from the nanopore genome data. Technical reasons for failure to obtain null mutants was thought be low because the same primers and system was used to generate the tagged lines. We have included this point in the discussion.

2. Gene Tagging

2.1. Cell cycle dependent localisations for some kinases, which are mentioned in the text, should be supported by the inclusion of the relevant images.

Response: The relevant images can be found in supplementary data 2

2.2. There are issues with the annotated localisations in Supplementary Data 2:

The localisations don't always match what is visible in the pictures and they also don't match the terms shown in Figure 3b. A non-exhaustive list of examples:

A6-A9, nucleoplasmic localisation not evident

A11, AMPGK, CBPK1, LmxM.21.0853, flagellar cytoplasm fluorescence not visible

C2, only nucleoplasm seems fluorescent

CK1.4, ATM, no cytoplasmic fluorescence visible

LmxM.33.3020, nucleoplasm not fluorescent

MKK4, LUK6, no cytoplasmic and flagellar fluorescence visible

The distinctions between flagellar pocket / endocytic not always clear, also basal body/kinetoplast. The criteria for defining a localisation as “endomembrane” are not clear. How was “flagellar cytoplasm” defined and how was it distinguished from axoneme or PFR?

I would recommend checking these again, within the set and also against the “landmarks” reference paper. The simplest approach may be just to use the terms shown in Figure 3b consistently.

Response: As suggested by the reviewer, we have edited the figures and text to just use the “landmarks” reference paper. We have re-checked all localisations against the landmarks.

2.3. Cyan (DNA) and green fluorescence is hard to distinguish in the overlay, a combination for example of purple / green would be clearer (like in Fig 3c, which shows very nice examples of the different localisations).

Response: We would prefer to keep the supplemental figure with Cyan and Green fluorescence

2.4. Most cells show a very elongated morphology. Is it certain these images were taken from actively dividing mid-log cultures, not stationary cultures? This may affect observed localisation.

Response: The images were taken from mid-log cultures, although even in mid-log there are a mixture of cell types. We show representative images from a population of cells.

3. Phenotyping – fitness in vitro and in vivo

The analysis focuses on survival in different life cycle stages, in culture, in cells, in mice and in sand flies. The design is elegant, tracking populations from promastigotes continuously through to different end-points with relevance to the actual life cycle.

3.1. The following clarifications would be helpful:

- Why were only 154 gene deletions out of 162 included in pooled screens?

Response: We had some technical issues validating some of the gene deletion mutants, so these were generated/confirmed after the library screen was performed. Also, several lines did not recover from stabilate in time for the screen. These contributed to the overall assessment of required vs dispensable, but indeed some were missing from the pool.

- What is the significance of the Δmca control? What does its trajectory look like and did it correspond to expectations?

Response: see response to reviewer 1 The control corresponded to expectations in that it has a similar trajectory to that described in our previous study (Damianou et al., 2020 Plos Pathogens)

- Were barcoded control lines included (as for sand fly infections) and what does their trajectories look like?

Response: No, the bar-coded control lines were generated after the in vivo main pooled library screen had been completed. This was to provide a comparison with the sand fly infection experiments carried out by Beneke et al (Plos Pathogens 2019), where the same type of controls were used.

3.2. For the fly infections, mutants were separated into three pools based on growth rate. Why, and was this also done for the other pooled screens? It is striking that there doesn't seem to be a "cluster" of mutants with reduced fitness in promastigotes but average or enhanced fitness in amastigotes. Could it be an intrinsic limitation of the assay that such mutants were systematically missed?

Response: Please see response to reviewer 1. There are a number of mutants that have a slow growth phenotype as promastigotes and have a relative loss of fitness (for example AMPKa, MPK6, CKA2 and TOR3). It is true that these mutants are poorly represented in the metacyclic population and therefore will be at a disadvantage in the macrophage and mice infection experiments. It would certainly be interesting to repeat the infection experiments with a pool of slow growers – this may reveal relatively enhanced fitness in amastigotes

4. The role of kinases in specific processes

Differentiation

4.1. In the title, abstract & introduction, there is a strong focus on differentiation, yet the paper contains only limited data that speaks directly to differentiation. (Selection of metacyclics and the detailed examination of $\Delta rdk1$, below). Under the Heading "Protein kinases important for differentiation from promastigotes to amastigotes", the paper reports 29 kinases important for amastigotes. This is clearly a very important finding, not least for drug discovery campaigns. It does not tell whether differentiation or other processes were affected. The title of the paper ("regulators of differentiation") doesn't accurately reflect the key findings.

Response: This is a valid point – the screen does not distinguish between those mutants that have a significant loss of fitness during differentiation itself and those that differentiate normally and then have a loss of fitness growing as amastigotes. We have added some text to the discussion to make this clear. We have also change the title to "Systematic functional analysis of Leishmania protein kinases identifies regulators of differentiation or survival."

4.2. For $\Delta rdk1$, the data provide strong evidence supporting a critical role of $rdk1$ in amastigotes. But do these mutants really have accelerated differentiation from amastigotes to promastigotes?

The HASPB blot which is presented as evidence supports this to some extent (although the unequal loading of the lanes raise the question how reproducible this was). I would be interested to know whether this acceleration was evident in an accelerated morphological change, which provides independent and easily measured markers (elongation of cell body and outgrowth of the flagellum).

Response: We have carried out the differentiation experiment as suggested and measured elongation of the cell body and outgrowth of the flagellum. The data do not support an accelerated differentiation from amastigotes to promastigotes. We feel this needs more investigation, so we have removed old figure S3d and the associated text in the results section saying that RDK1 is a repressor of differentiation. We have kept the remaining RDK1 data as validation for the screen.

A growth curve for axenic amastigotes (KO, WT and add-back) would show whether slower replication of $\Delta rdk1$ amastigotes could explain the lower parasite numbers in macrophages and mice, rather than a differentiation defect.

Response: We have carried out the growth curve and have included the data in a new supplementary figure 3d. This shows no significant difference between the cas9T7 line and $\Delta rdk1$.

Line 148, “identifies RDK1 as a Leishmania repressor of differentiation” – it is unclear what differentiation is being repressed (presumably amastigote to promastigote) and how that links to the observed defects in amastigotes.

Response: We have adjusted the text in light of responses above

4.3 Motility. An innovative trans-well assay was used to identify motility mutants from mutant pools. Further clarification on the details and validation would help to assess the power and limitations of this assay:

- Were all mutants included and is the full bar seq data available? In the Supplementary Table I can only see the “significant motility outputs” but not the full dataset.

Response: We actually screened both Pool1 and Pool 2 for motility defects. The full bar-seq dataset for Pool 1 had been included in Tab 1 of the Supplementary Table 1. We have now made two new tabs in the spreadsheet, which includes the bar-seq data for both screens. We have also edited the results section to explain this more clearly (and to address the other points below).

- What controls were done to validate this assay? Were any known paralysed, slow and fast swimming mutants tested to assess the discriminatory power of the assay?

Response: No controls were used to validate the assay – we wanted to carry out an unbiased screen.

- Mutants with a very short or absent flagellum are easily recognised just by microscopic examination of the culture. Was this recorded for the mutants, which were generated as individual cell lines, and how did the results match up with the trans-well assay?

Response: Microscopic examination of the mutants was not recorded systematically when the library was generated (but we probably should have done that). The motility screen was devised subsequent to the making of the library.

- Wiese's group previously reported that deletion of MPK3 and MKK1 resulted in very short flagella (refs 12 and 13) and reduced motility. Were these mutants present in this screen and could the authors comment on why they might have been missed in the trans well assay? Similar for PKA which was reported to have reduced motility (ref 42).

Response: MPK3 and MKK1 were present in the screen. MPK3 was identified as having a motility defect in our Pool 2 screen, but not MKK1. The PKA reported to have a motility defect in ref 42 was a heterozygote, not a null mutant (PKAC1) – in our screen this is an essential gene and so was not included in the motility screen.

5. Data accessibility

5.1. The supplementary Excel files contain very valuable data but should be properly labelled, with legends containing the minimal information to allow easy understanding of content, specifically Table legends stating what each Tab of Tables 1 and 2 contain, and supplementary data should be referenced in the text (e.g. morphology measurements for the identification of motility mutants).

Response: We have added legends to provide the additional information requested

5.2. While it is stated that “additional images are available [...] upon request”, it would be preferable to supply these additional images through a suitable repository.

Response: We have deleted this statement. Some additional images have been added (see response to reviewer 1)

6. Minor comments & typos

6.1 S5 Figure, the μ in the x Axis label doesn't show up properly.

6.2. 178, flagella (not flagellar)

6.3. line 348, do the authors mean: ...leading to fewer infective metacyclic forms (rather than

“less infective metacyclic forms”)?

6.4. Fig 6a Y-axis label, should presumably be the other way round (Flagella : Cell body ratio, as in S6 a)

6.5. Kinase orthology: Please specify how orthology was established and specify more precisely what “Conserved across everything (123)” means (conserved across surveyed kinetoplastids?).

6.6. Supplementary Figure 4 – the legend refers to mutants highlighted in red but the image is in black and white.

Response: We have addressed all the minor comments and typos

Reviewer comments, second round:

Reviewer #1 (Remarks to the Author):

The authors have addressed almost all comments satisfactorily, and I recommend the paper be accepted for publication.

There is however one point, however, (Point 5 in my initial review), that would deserve another comment: The comment and rebuttal are pasted here:

"5. Survival in the sandfly: The rationale for splitting the mutant set into three subpools according to growth rate at the promastigote stage (line 265) makes sense, as this stage is replicative in the sand fly --this reviewer assumes this strategy is designed to minimise competition from fast-growing parasites that would skew the data (if this is right, this may deserve a brief explanation after line 265). However, first, the growth rate may be different in the fly's midgut than it is under laboratory culture conditions. Second, the natural infective form (from the vertebrate host to the sandfly) is the amastigote. It appears from the methods description (lines 463-479) that flies were fed artificially with blood spiked with cultured promastigotes. Would this strategy miss kinases required for natural infection from the amastigote stage? A comment to this effect would be useful.

Response: We split the mutant set into three promastigote pools for practical reasons, as much as for technical reasons, as the pools were shipped from York to Prague and then used to infect sand flies immediately. Promastigote-initiated infections are a standard for all experiments done with *Leishmania* mutants. In addition, Sadlova et al. (2017) studied to what extent promastigote-initiated experimental infections differ from those initiated with amastigotes. Both groups developed heavy late-stage infections with the same localization, uniform representation of infective metacyclic forms and equal efficiency of transmission. Whilst in our present study some difference between promastigote and amastigote derived infections might be expected on day 1, the final outcome of *Leishmania* development on day 8 would be the same."

I understand the logistical problem that infecting flies with macrophages/amastigotes would raise; however, I do not think the response really addresses the point. Feeding flies with promastigotes will not provide any insights into defects in the amastigote-to-promastigote development stage transition. If there is a kinase that is essential for this transition, but not for promastigote survival in the fly, it will not be detected by feeding mutant promastigotes to the fly.

I would suggest a cautionary statement to this effect is added to the discussion. This does not require a new round of review and can be assessed by the Editor (if he/she concurs with me on that point!).

There are a few typos (for example the "gamma" symbols are replaced with a square, presumably during conversion to PDF), but these can be fixed at the production stage.

I really enjoyed assessing this manuscript and I have no doubt it will be very well received by the Tryp community and beyond.

Reviewer #3 (Remarks to the Author):

The revised manuscript addresses my previous questions.

One final query about Data Availability:

The revised manuscript says: "Whole genome sequencing data for *L. mexicana* Cas9T7 and gene deletion mutants are available from the European Nucleotide Archive under study accession

PRJEB27113.”

But, ENA Project: PRJEB27113 retrieves a study entitled: Sequencing of genomic DNA from three Streptomyces strains.

If this a typo, please could it be corrected.

Please find below a response to the reviewers' comments. Changes to the manuscript have been made in yellow highlight. Editorial changes have not been highlighted.

Reviewer #1 (Remarks to the Author):

I understand the logistical problem that infecting flies with macrophages/amastigotes would raise; however, I do not think the response really addresses the point. Feeding flies with promastigotes will not provide any insights into defects in the amastigote-to-promastigote development stage transition. If there is a kinase that is essential for this transition, but not for promastigote survival in the fly, it will not be detected by feeding mutant promastigotes to the fly.

I would suggest a cautionary statement to this effect is added to the discussion. This does not require a new round of review and can be assessed by the Editor (if he/she concurs with me on that point!).

Response: This is a very good point that we should have included in the revision. We have now added a statement into the discussion to address this point "Fifteen protein kinases were identified as being required for successful colonisation of the sand fly midgut. As sand flies were fed with procyclic promastigotes the phenotype investigated was establishment of infection in the insect host; the natural infection of sand flies with amastigotes and the role of protein kinases in differentiation of amastigote to promastigote was not investigated in this study. Development of Leishmania in the sand fly is far more complex than that observed in in vitro tissue culture and involves sequential differentiation from procyclic promastigote to the nectomonad form and then to the leptomonad form. (new text in yellow)

There are a few typos (for example the "gamma" symbols are replaced with a square, presumably during conversion to PDF), but these can be fixed at the production stage.

Response: We have corrected the formatting problem with the symbols

Reviewer #3 (Remarks to the Author):

The revised manuscript says: "Whole genome sequencing data for *L. mexicana* Cas9T7 and gene deletion mutants are available from the European Nucleotide Archive under study accession PRJEB271113." But, ENA Project: PRJEB271113 retrieves a study entitled: Sequencing of genomic DNA from three *Streptomyces* strains.

If this a typo, please could it be corrected.

Response: Apologies, this was a typo and has been corrected to PRJEB40373